# A central arctic extreme aerosol event triggered by a warm air-mass intrusion

Lubna Dada [1,12] ✉, Hélène Angot [1], Ivo Beck[1], Andrea Baccarini[1], Lauriane L. J. Quéléver[2], Matthew Boyer[2], Tiia Laurila[2], Zoé Brasseur [2], Gina Jozef[3,4,5], Gijs de Boer [3,6,7], Matthew D. Shupe [3,6], Silvia Henning [8], Silvia Bucci [9], Marina Dütsch [9], Andreas Stohl [9], Tuukka Petäjä [2], Kaspar R. Daellenbach [10], Tuija Jokinen[2,11] & Julia Schmale [1] ✉

Frequency and intensity of warm and moist air-mass intrusions into the Arctic have increased over the past decades and have been related to sea ice melt. During our year-long expedition in the remote central Arctic Ocean, a record-breaking increase in temperature, moisture and downwelling-longwave radiation was observed in mid-April 2020, during an air-mass intrusion carrying air pollutants from northern Eurasia. The two-day intrusion, caused drastic changes in the aerosol size distribution, chemical composition and particle hygroscopicity. Here we show how the intrusion transformed the Arctic from a remote low-particle environment to an area comparable to a central-European urban setting. Additionally, the intrusion resulted in an explosive increase in cloud condensation nuclei, which can have direct effects on Arctic clouds' radiation, their precipitation patterns, and their lifetime. Thus, unless prompt actions to significantly reduce emissions in the source regions are taken, such intrusion events are expected to continue to affect the Arctic climate.

The Arctic is warming at a rate roughly thrice as fast as the rest of the globe due to Arctic amplification[1,2]. While its detailed causes remain to be quantified[3,4], a large number of studies attributed the amplification to anomalous poleward atmospheric transport in the form of warm air-mass intrusions[5–8]. Air-mass intrusions arriving from mid-latitudes can introduce moisture and perturb Arctic temperatures. The frequency and intensity of warm and moist air-mass intrusions control the inter-annual variability in Arctic mean surface air temperature, humidity and downward longwave radiation[9,10]. As warm air-mass intrusions arrive in the Arctic at varying altitudes, boundary layer characteristics and cloud properties are affected, albeit not uniformly[11–13]. A number of studies have associated melting processes on Greenland's ice sheet and changes in sea ice concentration to intense warming events introduced into the Arctic during springtime[6,12,14,15], highlighting the importance of such events.

Synoptic-scale warm, and moist, air-mass intrusions into the Arctic represent extreme, intense and anomalous short-lived events (temperatures close to or above 0 °C, lasting for between 1 and

[1]Extreme Environments Research Laboratory, Ecole Polytechnique Fédérale de Lausanne (EPFL) Valais Wallis, 1951 Sion, Switzerland. [2]Institute for Atmospheric and Earth System Research, INAR/Physics, Faculty of Science, University of Helsinki, 00014 Helsinki, Finland. [3]Cooperative Institute for Research in Environmental Science, University of Colorado, Boulder, CO 80309, USA. [4]National Snow and Ice Data Center, University of Colorado, Boulder, CO 80309, USA. [5]Department of Atmospheric and Oceanic Sciences, University of Colorado Boulder, Boulder, CO, USA. [6]Physical Sciences Laboratory, National Oceanic and Atmospheric Administration, Boulder, CO 80305, USA. [7]Integrated Remote and In Situ Sensing, University of Colorado, Boulder, CO 80309, USA. [8]Leibniz Institute for Tropospheric Research, Permoserstrasse 15, 04318 Leipzig, Germany. [9]Department of Meteorology and Geophysics, University of Vienna, 1090 Vienna, Austria. [10]Laboratory of Atmospheric Chemistry, Paul Scherrer Institute, 5232 Villigen, Switzerland. [11]Climate & Atmosphere Research Centre (CARE-C), the Cyprus Institute, P.O. Box 27456 Nicosia 1645, Cyprus. [12]Present address: Laboratory of Atmospheric Chemistry, Paul Scherrer Institute, 5232 Villigen, Switzerland. ✉e-mail: lubna.dada@epfl.ch; julia.schmale@epfl.ch

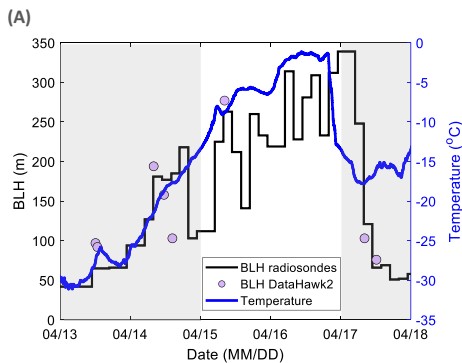

(A)

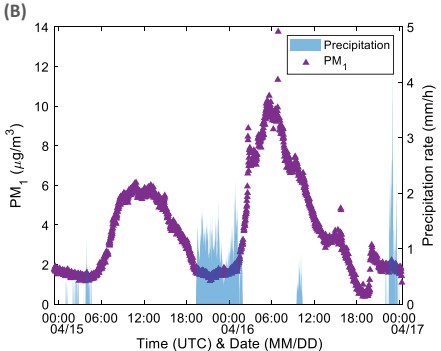

(B)

**Fig. 1 | Boundary layer height, temperature and particulate matter during the warm air-mass intrusion event. A** Evolution of the boundary layer height (BLH, left axis, solid black line and lilac markers) and temperature (right axis, solid blue line), the unshaded area represents the warm air-mass intrusion event and is the focus of this study. **B** Particulate matter with diameter smaller than 1 μm (PM$_1$, left axis, purple triangles) and precipitation rate (right axis, shaded area), the two peaks of the warm intrusion event are separated by precipitation.

3 days)[16] related to blocking situations of the large-scale circulation[7,17]. In the last decade, such events have been more frequent than before and their duration longer[16]. In fact, the number of moisture intrusions associated with increased surface temperature in the Arctic region during December and January have approximately doubled since the 1990s[18]. Based on model projections of future climate, they are predicted to further increase in frequency and duration[17]. While warm, moist air-mass intrusions into the Arctic are typical in both winter and spring[8], most research has focused on winter[19] with only little attention on transition seasons. Yet, springtime intrusions are of high importance because they have direct connection to the speed of sea ice retreat through triggering early and short episodes of melt, which lower the surface albedo[17,20].

As part of the Arctic atmosphere, local aerosol concentrations, their chemical composition and properties depend on atmospheric circulation and are affected by season, local emissions and emissions far away that are atmospherically processed during long-range transport. Thus, the aerosols and their properties are expected to be largely affected by warm and moist intrusion events. Yet, due to measurement limitations, the impact of such intrusions on the aerosol load and related climate impacts is not well understood nor quantified.

For instance, measurements of aerosol optical properties in the Arctic provide high spatial and long-term climatological information important for model simulations, however remain mainly site-specific and do not often extend fully to the central Arctic Ocean[21–23]. Additionally, while land-based observatories, provide climatological information about Arctic particle size distribution and mass composition that is highly valuable for understanding Arctic aerosol processes and for modeling their climate effects[24–26], they are not necessarily representative for aerosol characteristics and impacts over the remote central Arctic Ocean. The effects of warm air-mass intrusions on the particle population have not yet been studied over the central Arctic Ocean, because in-situ measurements were previously only conducted in summer[27,28].

The year-long MOSAiC expedition (Multidisciplinary drifting Observatory for the Study of Arctic Climate; Oct. 2019-Sept. 2020) enabled scientists to study the central Arctic's atmosphere in detail[29,30]. In September 2019, the German research icebreaker Polarstern set sail from Tromsø, Norway, to spend a year drifting through the Arctic Ocean, trapped in ice, to closely observe all the pieces of the Arctic climate puzzle, including the atmosphere, sea ice, ocean, ecosystem, and biogeochemical processes.

During the expedition, a record minimum sea ice extent was measured in July 2020 (compared to 1979-2020)[20]. This major sea ice retreat was preceded by an extreme air-mass intrusion event in April 2020, which was high in moisture, temperature and longwave

radiation re-emitted by low level clouds in the Arctic, all being the highest in the last 40 years[10].

In this study, we focus on the effects of this record-breaking warm and moist air-mass intrusion event on the central Arctic aerosol population and cloud properties, and therefore on the central Arctic Ocean climate. Our unique, in-situ, measurements in the remote central Arctic Ocean demonstrate the capability of pollutants emitted in lower latitudes to drastically alter the Arctic atmosphere, one of the world's most climate sensitive locations.

## Results

### Air-mass intrusion into the central Arctic Ocean

Along the drift trajectory of Polarstern in 2020, an anomalously fast increase in near-surface air temperature from −30.8 °C on April 14 to −3.1 °C on April 16 was observed (Supplementary Fig. 1). The increase in temperature was associated with an increase in water vapor mixing ratio ($w$) and constituted a warm and moist air-mass intrusion into the central Arctic Ocean. The periods of focus in this study are 2nd–3rd of April (background) and 15th–16th of April (intrusion), representing contrasting periods in terms of temperature and moisture evolution, and are shown in Supplementary Fig. 1.

During our observation period, warm and moist air was observed intruding from Eastern Europe over the Barents Sea towards the central Arctic over a large region including the Polarstern position (Supplementary Figs. 2–3). Over the two days of the warm air advection, in addition to the steady increase in near-surface air temperature, we also observe a lift in the boundary layer height (BLH) (Fig. 1A), which likely facilitated mixing in the part of the intrusion air-mass transported at <1000 m. Two peaks in PM$_1$ (particulate matter with diameters smaller than 1 μm, approximated from non-refractory particulate matter and black carbon (BC)), separated by a dip, were observed, with the second peak higher in magnitude. The difference in mass concentration between the two peaks suggests advection of differently polluted air-masses as well as different vertical mixing, possibly combined with more or less wash out during transport, and/or more or less secondary processing. The two peaks are separated by a local precipitation event (Fig. 1B).

During the background period, the footprint of emission sensitivity derived from the Lagrangian transport model FLEXPART[31] suggests that the air originated from the Kara Sea region and circulated around the North Pole (Fig. 2A left, Supplementary Fig. 4–5). Along the transport pathways, pollutants age and are processed, resulting in widespread and well-characterized Arctic haze[32,33]. In contrast, during the first day of the warm air-mass intrusion period, FLEXPART emission sensitivity shows direct and quick transport across the Barents Sea from the Nenets Autonomous District and the city of Vorkuta in Russia

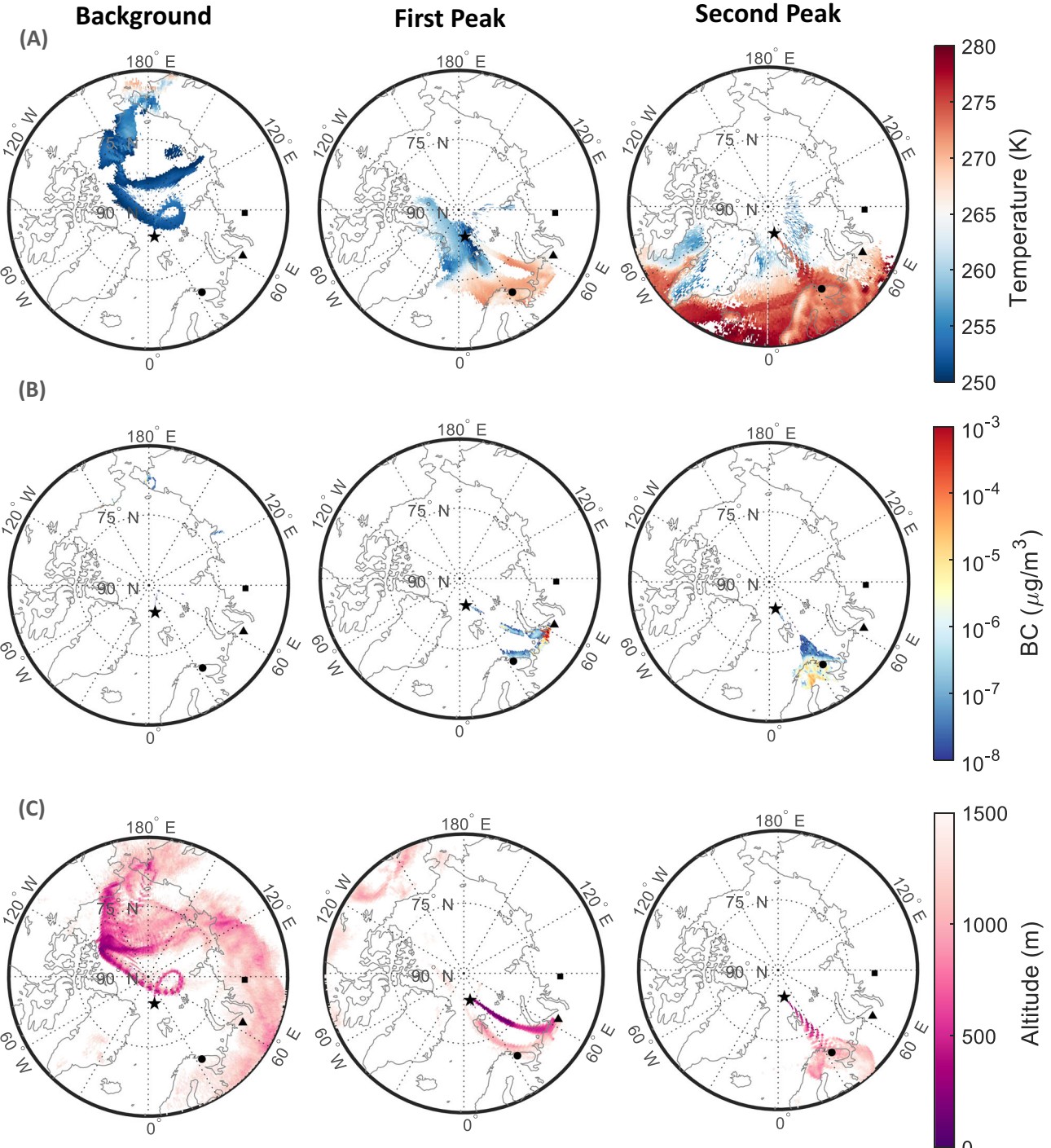

**Fig. 2 | Source area, temperature, air-mass trajectories and trajectory height.**
**A** Mean temperature of the air-mass with simulated particles residing below 100 m altitude, obtained from 7-day backward simulations with FLEXPART during the background period (left panel), first peak of the intrusion event (middle panel) and second peak of the intrusion event (right panel). **B** Black carbon source contribution (below 100 m a.g.l.), as a surrogate for anthropogenic pollution, for a passive air tracer obtained from 7-day backward simulations with FLEXPART during the background period (left panel), first peak of the intrusion event (middle panel) and second peak of the intrusion event (right panel). Inserts of **B** are shown in Fig.S5.

**C** Average altitude of all particles residing below 1500 m, from the 7-day FLEXPART backward calculation for the background period (left panel), first peak of the intrusion event (middle panel) and second peak of the intrusion event (right panel). The structures that resemble "ripples" are due to the time resolution of the model output, three hours. The position of Polarstern is marked with a star, that of Norilsk (69.3558° N, 88.1893° E) with a square, that of Vorkuta (67.4969° N, 64.0602° E) with a triangle and that of Murmansk (68.9733° N, 33.0856° E) with a circle. Polarstern drifted from 84.95° N, 14.99° E on 02.04.2020 at 12:00 to 84.34° N, 13.09° E on 16.04.2020 at 12:00.

(Fig. 2A-B middle). This region is a known source region of Arctic air pollution from gas flaring[34]. During the second day, the FLEXPART emission sensitivity footprints were high over northern Finland and the Kola Peninsula in Russia, near Murmansk, another very polluted

region, known for its metallurgical industry (Fig. 2A, B right, Supplementary Figs. 5–6). Air-masses transported via such "intrusion pathways" from lower latitudes to the Arctic extend over thousands of kilometers, and occur over only several hours to a few days[2], but are

nevertheless subject to substantial atmospheric chemical and microphysical transformation. For instance, based on the trajectory analysis, the air-masses arriving to Polarstern on April 16th spent less than two days during their travel from the Kola Peninsula in Russia. During the intrusion period, transport occurred consistently at low-level via the Barents Sea (Fig. 2C), which is known to be the most efficient transport pathway of pollutants into the Arctic from continental source regions in northern Eurasia[33].

## Aerosol size distribution, number and mass concentrations

In terms of aerosol number size distribution, the background period is representative of Arctic haze conditions (Fig. 3A). The accumulation mode (100–1000 nm) is very stable at around 150 cm$^{-3}$ (Fig. 3C). During the intrusion period, the particle number increased drastically and multiple diameter modes appeared (Fig. 3B). Two distinct peaks appear during the intrusion period with a peak accumulation mode concentration of 300 and 700 cm$^{-3}$ during the first and the second part, respectively (Fig. 3D). A substantial number of particles is also observed in the Aitken mode (25–100 nm) with a peak concentration of 450 and 600 cm$^{-3}$ during the first and second peak, respectively (Fig. 3D). While the background accumulation mode concentration is comparable to median April concentrations measured at land-based Arctic locations including mount Zeppelin (Svalbard), Villum research station at Station Nord (VRS; Greenland), Alert (Canada), Utqiagvik (Alaska) and Tiksi (Russia) (100–200 cm$^{-3}$)[25,35], concentrations during the intrusion period are similar to summer average accumulation mode particle concentrations observed in rural[36] and even urban[37] mid-latitude locations. This comparison shows the dramatic impact of the intrusion on the aerosol number size distribution in the remote Arctic.

During April 2020, $PM_1$ varied between 0.002 and 13.8 μg/m$^3$ (Supplementary Fig. 7). During the background period (2nd–3rd of April), the $PM_1$ mass fluctuated between 1 and 2 μg/m$^3$ (Fig. 4A). This mass concentration is comparable to the average mass concentration observed in VRS (Greenland) between February and May 2015[38], Mace Head (Ireland) in spring 2012[39], and that observed on top of a mountain within the free troposphere at Jungfraujoch (Switzerland) in spring 2008[40]. During the warm air-mass intrusion, we observe a 4- and 6-fold increase in $PM_1$ mass concentration during the first and second peak, respectively (Fig. 4B). In fact, such a mass concentration (up to ~10.3 μg/m$^3$ with a time resolution of 15 min) is atypical for the central Arctic Ocean[41]. For comparison, $PM_1$ average springtime mass concentrations in central and southern Europe range between 5 and ~18 μg/m$^3$[39,40,42]. The mass concentration observed during the intrusion period is extreme within the central Arctic Ocean context and demonstrates the large effect of mid-latitude pollution transported to the Arctic.

## Aerosol chemical composition

At the same time, a strong variation in the mass composition of the aerosol was observed (Supplementary Fig. 7). Here, the contributions of the ship exhaust emissions are eliminated by applying the pollution mask (see Methods section, instrumentation subsection) to focus on the Arctic aerosol during background periods and the warm air intrusion event. During the background period, the 15 min average sulfate ($SO_4^{2-}$) concentration varied between 0.4 and 1.0 μg/m$^3$ (Fig. 4A). In the Arctic, in general, $SO_4^{2-}$ originates from sea-salt, dimethyl sulfide (DMS) and long-range transport[43]. Our observed winter/early spring concentrations are comparable to the concentrations observed at multiple Arctic land-based locations[24,44,45]. During this period, $SO_4^{2-}$ contributed ~ 50% of the total $PM_1$ which is typical for aged air-masses originally transported from Asia and Europe[41]. In comparison, during the warm air-mass intrusion period, the $SO_4^{2-}$ concentration exceeded 3.4 (58 %) and 4.2 μg/m$^3$ (44 %) during the first and second peak, respectively. Such high sulfate concentrations are

not typical of the Arctic[43], neither are they typical to any European location regardless of whether it is remote, background, marine or even urban[42]. This off-scale $SO_4^{2-}$ concentration cannot be attributed to sea-salt as the insignificant intensity of the NaCl fragments, NaCl$^+$ (m/z 58) and Cl$^-$ (m/z 37) as measured by the aerosol mass spectrometer on MOSAiC, did not vary, regardless of the substantial variation in the $SO_4^{2-}$ mass. This observation suggests that the majority of the $SO_4^{2-}$ mass results from secondary formation following $SO_2$ emission and long range transport (Supplementary Fig. 5)[46]. While some of the secondary $SO_4^{2-}$ could be attributed to DMS oxidation upon the travel of the air-mass above open ocean areas, the contribution in our case is expected to be minor at this time of the year[24,47,48]. The much higher concentration of $SO_4^{2-}$ observed in the central Arctic compared to European sites reflects the strong $SO_2$ sources in Russia (Supplementary Fig. 6). For instance, the Kola peninsula is notorious for extremely high $SO_2$ emissions from the metal smelters[49]. Although the transport was relatively fresh, <2 days old, $SO_2$ was completely converted to the particulate phase by the time it arrived at Polarstern (Supplementary Fig. 6). Once emitted, vapors undergo chemical transformation, aqueous phase processing and gas-to-particle partitioning during their travel to the central Arctic Ocean[38,50]. Interestingly, neither a noteworthy peak in $SO_4^{2-}$ nor in $SO_2$ was observed at Zeppelin, Svalbard during the intrusion event on April 15th–16th (Supplementary Fig. 5), although the location was affected by the intrusion in terms of temperature (Supplementary Fig. 2). It appears that while the advection of temperature occurs over a broad swath, the advection of aerosols/pollution happens over narrower parts of the overall event due to specific point sources embedded within the general source region. Such a limited observation of the intrusion in certain land-based Arctic locations (here, Zeppelin) and their absence at others (e.g., VRS, Greenland), demonstrates the importance of in-situ central Arctic Ocean measurements as land-based observations cannot be simply extrapolated.

Organic aerosol (Org), which is the second largest contributor to $PM_1$ in our case, varied in 15 min average concentrations between 0.4 and 0.8 μg/m$^3$ (41 %) during the background period (Fig. 4A). Our observations are consistent with the concentrations observed at VRS in Feb–May 2015[38], in which the Org concentration remained close to 1 μg/m$^3$ while contributing only to 24% of the total $PM_1$. During the intrusion period, the Org concentration exceeded 2 μg/m$^3$ (37 %) and 5 μg/m$^3$ (44%) during the first and second peak, respectively. Such high concentrations are untypical for the central Arctic and are even higher than those observed in coastal locations at Mace Head and Corsica, which are affected by marine organic sources[39]. The concentrations are comparable to Org concentration at a range of rural and urban locations in Europe[42]. Interestingly, the fractional contributions of the different mass components during the first peak of the intrusion (Fig. 4D) prior to the precipitation are different from those observed during the second peak (Fig. 4E). During the second peak, the fractional contribution of Org increased to 44% and that of $SO_4^{2-}$ decreased to 44% as shown in Fig. 4E. These changes follow most likely from differences in emissions in the source regions and transport trajectories.

Black Carbon, which is an indicator for direct particle emissions from combustion processes, shows a relative contribution of <5% throughout April (Fig. 4C–E). The BC concentration increases from ~0.1 μg/m$^3$ during the background period to ~0.5 μg/m$^3$ (15 min average) during the intrusion period, specifically during the 2nd peak, when gas flaring and metallurgical industry regions are part of the source mix (Figs. 2 and 4). The background BC concentration is typical of late winter-early spring Arctic concentrations[38,44]. While the intrusion concentrations, especially during the second peak, are exceedingly high for the central Arctic ocean, they are close in magnitude to sub-Arctic Kevo, in northern Finland during Spring[24,44] and to near-source high Arctic concentrations measured in October during the

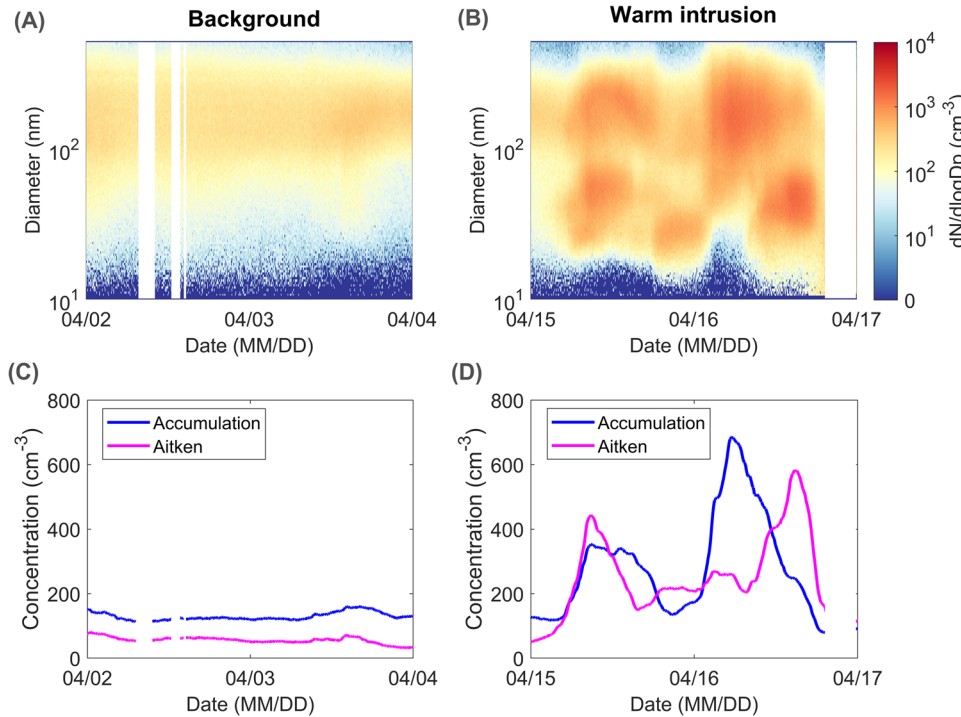

**Fig. 3 | Background and warm intrusion particle number size distribution.** Particle number size distribution during the (**A**) background period and (**B**) warm air-mass intrusion event. Accumulation mode (blue) and Aitken mode (magenta) particle number concentrations during the (**C**) background period and (**D**) warm air-mass intrusion event. Data points affected by ship exhaust emissions are removed.

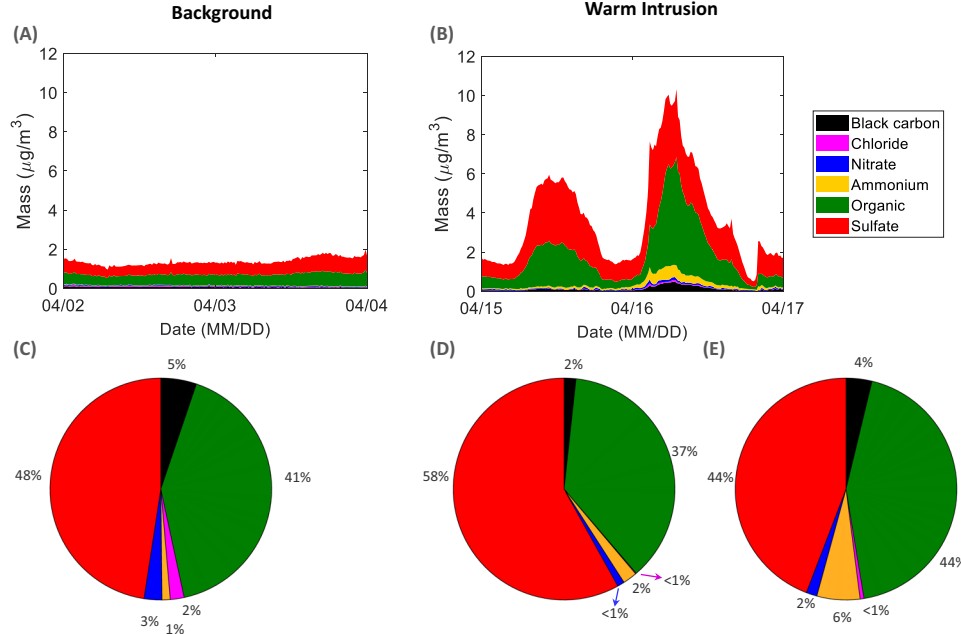

**Fig. 4 | Central Arctic aerosol mass composition.** Aerosol mass composition (averaged over 15 min) during (**A**) the background period and (**B**) the warm air-mass intrusion event. Pie charts show the percentage contribution of each of the chemical components to the total PM$_1$ (particulate matter with diameter smaller than 1 μm) during (**C**) the background period, (**D**) the first peak in PM$_1$ during the warm air-mass intrusion event (excluding precipitation periods) and (**E**) the second peak in PM$_1$ during the warm intrusion (excluding precipitation periods).

"Sever-2015" expedition[51]. During the background period, no clear connection between SO$_4^{2-}$ and BC is apparent (Supplementary Fig. 8A) likely as a result of their low concentrations but also suggesting a broad mix of sources that cannot be back-traced to individual emission regions or sources. Conversely, during the warm intrusion period, the correlation between the two species increases substantially with $R^2 = 0.44$ ($p$ value $= 3.8 \times 10^{-4}$) and $R^2 = 0.70$ ($p$ value $= 3.9 \times 10^{-7}$), during the 1st and 2nd peak, respectively. The high significant correlation is an indication of the common transport of BC and SO$_4^{2-}$ from the source region to the central Arctic. A higher BC-to-SO$_4^{2-}$ ratio during

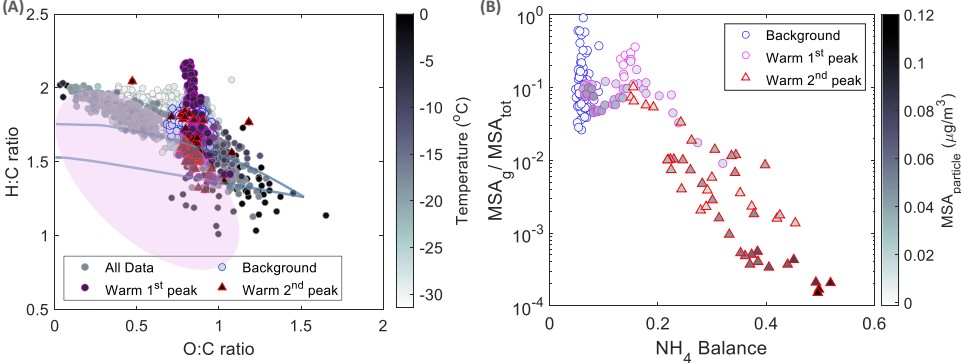

**Fig. 5 | Aerosol organic composition, acidity and MSA partitioning.**
**A** Relationship between the estimated hydrogen-to-carbon (H:C) and oxygen-to-carbon (O:C) ratios of organic species as Van Krevelen diagram. The ellipse represents data points below 300 m altitude in the summertime Canadian Arctic measured by Willis et al.[55] Blue lines are from Ng et al.[53], representative of mid- latitude ambient O:C and H:C ratios. "All data" refers to data points during April 2020. **B** Methane sulfonic acid (MSA) partitioning (measured gas phase to total MSA) as a function of aerosol acidity approximated by $NH_4^+$ balance colored with particle phase MSA measured with the HR-AMS.

the second peak, relative to the first peak (higher slope—Supplementary Fig. 8A), indicates stronger BC emissions relative to sulfur (here total sulfur = $SO_4^{2-}$ given that $SO_2$ concentrations are below detection limit). This could be attributed to the mix of sources at the emissions' origin.

Ammonium ($NH_4^+$) concentrations were expectedly small during the background period (0.01–0.03 µg/m³ (15-min averages); Fig. 4), while the concentration increased slightly with increasing temperature (Supplementary Fig. 9). During the air-mass intrusion period, $NH_4^+$ exceeded 0.7 µg/m³ (15 min average) and contributed 6% of the total $PM_1$ mass. $NH_4^+$ originating from ammonia ($NH_3$) arrives in the Arctic via long range transport of biomass burning and agricultural emissions[52]. The background concentrations are comparable to those observed between February and May in 2015 at VRS[38]. Similar to all other $PM_1$ components, particulate $NH_4^+$ concentrations during the intrusion period are not typical for the Arctic but rather resemble those observed at mid-latitude European rural sites[42]. Indeed, during the second peak of the intrusion, we find a high correlation between $NH_4^+$ and $SO_4^{2-}$ ($R^2 = 0.89$, $p$ value = $4.1 \times 10^{-12}$) and a slope of 0.16 (Supplementary Fig. 8B), demonstrating that $NH_4^+$ is transported in the particulate form as $(NH_4)HSO_4$ to the Arctic. During background conditions, the correlation between these two species is $R^2 = 0.43$ ($p$ value = $5.5 \times 10^{-4}$), showing again the widespread mix of sources building up to Arctic haze. From the change in particulate $NH_4^+$ concentrations follows a change in aerosol acidity. The aerosols' acidity and how it affects the Org fraction and specifically methane sulfonic acid (MSA) partitioning is described in detail in the following section.

**Changes in organic composition upon the intrusion event**
The Org fraction varies in composition between the background and warm air intrusion event (Fig. 4 and Supplementary Fig. 10). While the ratio of non-oxygenated to oxygenated families was similar between the background and intrusion periods, the degree of oxygenation varied substantially, with higher oxygenation during the intrusion particularly in the second peak. The average normalized spectra of each of the background, first peak and second peak of the intrusion are considerably different especially at higher m/z ratios (Supplementary Fig. 10). Specifically, the second peak of the intrusion shows a clear contribution of higher mass to charge ratios (m/z > 45 amu) to the total mass and a higher degree of oxygenation. This observation could be partially explained by different source regions, different oxidation mechanisms and heterogeneous processing. To further investigate organic aerosol ageing mechanisms, variability of hydrogen to carbon (H:C) and oxygen to carbon (O:C) ratios are shown as a Van Krevelen diagram in Fig. 5A. We observe a clear anti-correlation between the O:C

and H:C ratios for all of April within Arctic air-masses. H:C and O:C ratios outside of the intrusion period feature a slope of −0.5, indicative of both the addition of acid and alcohol/peroxide functional groups without fragmentation, and/or the addition of acid groups with simultaneous fragmentation[53]. During the first peak of the warm intrusion period, coinciding with relatively fresh air-masses (<2 days), high H:C ratios were observed. The H:C ratio decreases with the evolution of the intrusion period from more than 2.2 to less than 1.5, coinciding with the increasing temperature and aerosol acidity (Supplementary Fig. 12), as discussed in more detail in the next section. These data points, during the intrusion period, however, hold a distinct stable O:C ratio around 0.8, representing highly oxygenated organic aerosol[54].

Altogether, the different aerosol composition and H:C and O:C ratios during the background and intrusion period represent distinct emission source types, partitioning mechanisms and atmospheric aging processes including the differences between the two peaks of the intrusion event itself. Compared to the summer Canadian Arctic O:C and H:C range (ellipse in Fig. 5A)[40,55], the slope of our observations is less steep, i.e. for a given H:C ratio, our results show higher O:C ratio, indicating a higher degree of oxygenation. The higher oxygenation is likely explained by the longer atmospheric residence time of haze aerosol in our case compared to the potentially fresher emissions observed by Willis et al.[55] in Resolute Bay, within the Canadian Arctic Archipelago, as well as the strong oxygenation from heterogeneous cloud processing during the intrusion period. Compared to ratios expected from mid-latitudes[53] (solid lines in Fig. 5A) our values show higher H:C ratios and lower O:C ratios. Higher H:C ratios have been found representative of primary marine aerosol[55], which could particularly contribute to the background conditions in which long-range transported aerosol from the open ocean also accumulates in the central Arctic.

**Changes in aerosol acidity and MSA partitioning**
Aerosol acidity, here determined by the aerosol chemical composition, is a central factor in heterogeneous chemistry. We estimated the aerosol acidity to better understand its influence on the Org aerosol and on gas-to-particle partitioning especially of local marine compounds (here, MSA as a surrogate). Generally, aerosol acidity is primarily defined by the extent to which $SO_4^{2-}$ aerosol is neutralized by $NH_4^+$, producing ammonium bisulfate, or even ammonium sulfate. Previous model studies have shown predominantly acidic submicron aerosol throughout the depth of the Arctic troposphere in spring[52]. Here, we used the $NH_4^+$ balance calculation (see methods, subsection determination of aerosol acidity) and the Extended Aerosol Inorganic

Model (E-AIM)[56] to estimate the aerosol acidity during our measurement period based on the in-situ AMS measurements (Supplementary Fig. 11). An increased $NH_4^+$ balance reflects a higher neutralization capacity, and therefore a decreased acidity. The E-AIM model on the other hand, provides a direct estimate of the aerosol's pH[56]. Both approaches agree with each other, and show that the general state of the Arctic aerosol is acidic (Supplementary Fig. 11). The most acidic aerosol (pH ~ −1) are those measured during the background period (Supplementary Fig. 11A). The least acidic aerosol (pH ~ 0.5), although still very acidic, are observed during the second peak of the warm air-mass intrusion period, associated with peak $NH_4^+$ concentrations arriving to the Arctic (Supplementary Fig. 11A). Our observations are in line with those reported for clean/remote regions during the background period and those reported for remote polluted areas during the warm intrusion period[57].

We note a clear negative dependency of the H:C ratio on aerosol acidity during the intrusion period, suggesting that changes in Org's composition might be linked to aerosol acidity (Supplementary Fig. 12). To go further into the effect of aerosol acidity on the aerosol organic fraction, we select an organic molecule relevant for marine and Polar locations, MSA, which is produced from aqueous and gas phase oxidation of DMS, derived from dimethylsulfoniopropionate, a gas produced by phytoplankton[58] that participates in marine secondary aerosol formation[47,48,59,60]. MSA partitioning is mediated by gas-phase concentration, temperature, relative humidity and aerosol composition and properties[60]. To understand the effect of the warm air-mass intrusion together with changes in aerosol composition and acidity on MSA partitioning, we examine gas and particulate phase concentrations and variability. Gaseous MSA concentration was roughly $1 \times 10^6$ molecules.cm$^{-3}$ ($1 \times 10^{-4}$ µg/m³) in the first half of April and increased to ~$5 \times 10^6$ molecules.cm$^{-3}$ ($5 \times 10^{-4}$ µg/m³) in the second half of April, excluding the intrusion period (Supplementary Fig. 11B). This increase is related to increasing air temperature, sea ice melt, and increased radiation, biological activity and DMS emissions. These measured concentrations are similar to those observed at Ny-Ålesund (Svalbard) in April 2017, slightly higher than concentrations in VRS (Greenland) in April 2015, and more than one order of magnitude higher than those observed in summer-autumn 2017 at VRS and in September 2018 in the central Arctic[48], all observed with the same type of instrument. An increasing MSA concentration in spring is typical of its seasonal cycle. However, during the intrusion period, we observe a sharp increase in half-hourly gas-phase MSA concentrations exceeding $2.5 \times 10^7$ molecules.cm$^{-3}$ ($4 \times 10^{-3}$ µg/m³) during the first peak and, interestingly, completely depleted close to $1 \times 10^5$ molecules.cm$^{-3}$ ($1.7 \times 10^{-5}$ µg/m³) during the second peak (Supplementary Fig. 11B). This depletion is accompanied by a sharp increase of particulate MSA reaching 0.11 µg/m³, compared to detection limit concentrations (0.019 µg/m³) during the background period, and a maximum of 0.042 µg/m³ during the first peak (Supplementary Fig. 11B). The ratio of gas to particle phase MSA shows a substantial dip during the second peak reaching $1.5 \times 10^{-4}$ in comparison to the rest of the month when the ratio was larger than 0.1, showing near-complete partitioning of MSA into the particle phase. The ratio of gas phase MSA partitioning into the particle phase is hence fundamentally different between the two peaks. A similar result is retrieved from E-AIM simulations (Supplementary Fig. 11B). The model shows a clear dependency of the MSA partitioning ratio on aerosol acidity (Supplementary Fig. 11B). Less acidic aerosol (higher $NH_4^+$ balance) is related to less gaseous phase MSA, and is accompanied by an increase in particle phase MSA, hence an increased partitioning (Fig. 5B). In fact, according to the E-AIM results, the water content was substantially higher during the second peak compared to the first, which would also have a direct effect on MSA partitioning. Such a result highlights the role of warm intrusion events in modifying not only aerosol composition but also gas phase composition−hence atmospheric chemistry in the central Arctic. With

future expected decreases in $SO_2$ emissions and anticipated increase in $NH_3$ emissions[61], the resulting less acidic particles could lead to substantial changes in organic aerosol constituents, such as increased partitioning of MSA into the particle phase.

## Impact of the warm air mass intrusion event on clouds

In the climatically sensitive central Arctic, changes in cloud condensation nuclei (CCN) number and properties can have significant effects on cloud properties and radiative balance[62]. CCN observations in the Arctic region are minimal[35,39], and almost absent in the central Arctic[55,59]. During our observation period in April 2020, CCN number concentrations varied insignificantly within the supersaturation (SS) range of 0.18–0.78% during the background period (Fig. 5A–C). The measured CCN number concentration across all SS (100–200 cm$^{-3}$) (Fig. 6A) is typical for the springtime lower Arctic troposphere, as observed at Ny-Ålesund (Svalbard) over a period of 7 years between 2007 and 2013[35] and in Utqiagvik (Alaska) in 2007-2008[39]. During the intrusion event, CCN number concentrations drastically increased for all SS to the highest values measured during the expedition (Fig. 6B and Supplementary Fig. 13).

The median CCN number concentration at 0.29% SS (371.2 cm$^{-3}$) on warm intrusion days, excluding precipitation periods, was 4.2 times higher than the median (88.4 cm$^{-3}$) and almost three times higher than the 75th percentile (130.0 cm$^{-3}$) of all MOSAiC days (Fig. 6C). In general, the warm intrusion CCN number concentration lies above the 98th percentile of CCN concentrations observed for the entire year of MOSAiC (Fig. 6C). This follows from the off-scale median particle number concentration (695.6 cm$^{-3}$) during the warm intrusion, excluding precipitation periods, which is 5.2 times higher than the median (133.5 cm$^{-3}$) and 2.8 times higher than the 75th percentile (244.4 cm$^{-3}$) of all MOSAiC days. The warm intrusion particle number concentration lies within the 95th percentile of all expedition data (Fig. 6D). This simple statistical evaluation shows the extreme effect of the warm air-mass intrusion on the CCN population.

Besides high CCN number concentrations, CCN characteristics in terms of activation ratio, critical diameter and hygroscopicity were clearly distinct between the background and intrusion periods. A more detailed analysis of CCN characteristics during background and intrusion events is shown in Supplementary Fig. 14. Activation ratios (relative to the particle population >30 nm) are higher during background conditions for all SS. This would normally point towards a higher hygroscopicity of the background aerosol. However, given the starkly different size distributions, we investigated activation diameter at each SS for both time periods based on the particle number size distribution and CCN number concentration. We see (Supplementary Fig. 14C) that the critical diameter for SS = 0.78 % varies between 40 and 60 nm for background conditions, while it mostly oscillates around 40 nm during the first peak of the intrusion (Supplementary Fig. 14D). Hence, a more hygroscopic aerosol during the first peak of the intrusion is expected. During the first peak of the intrusion there is relatively more particulate sulfate compared to background conditions, explaining the higher hygroscopicity, with the remaining mass essentially contributed by organics, which are also more oxygenated, hence more hygroscopic, compared to the haze situation. During the second peak of the intrusion, contributions of less hygroscopic compounds such as BC and organics, explain the increase of the critical diameter to above 60 nm around 6:00 UTC on 16th of April.

To confirm our hypothesis, Supplementary Fig. 14 shows the hygroscopicity parameter (kappa) derived from aerosols' chemical composition (see Methods; subsection determination of aerosol hygroscopicity) during the background and intrusion event period. We find that, during the background period and the first peak of the intrusion, there is a small variability in hygroscopicity (Supplementary Fig. 14G–H), with only slightly higher kappa values during the first intrusion period (~0.58) than the background (~0.54) explained by the

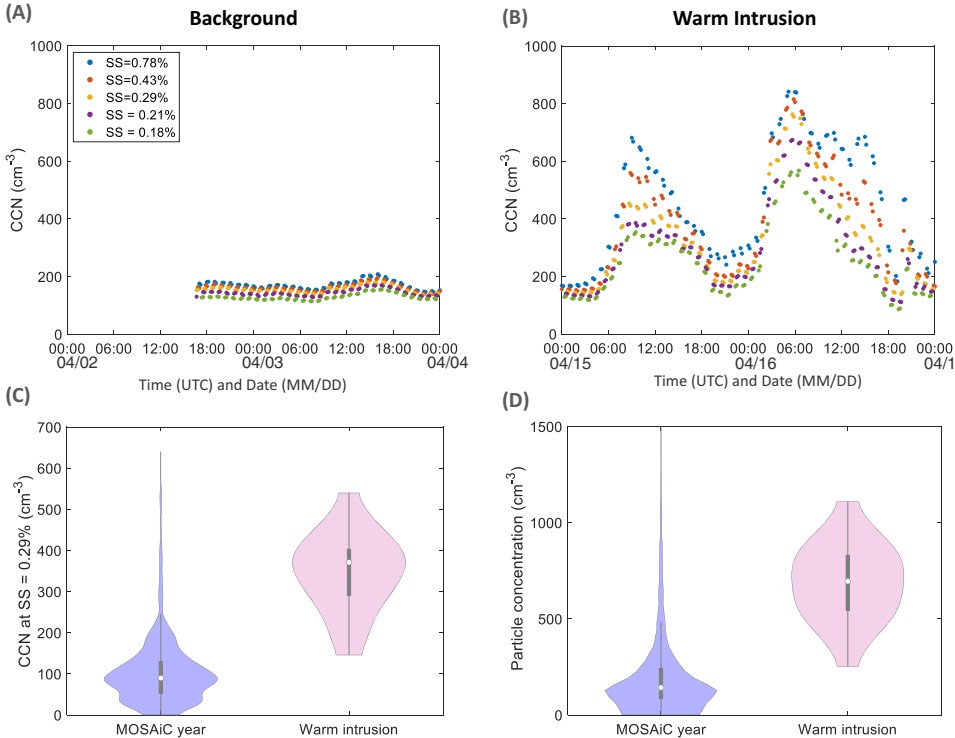

**Fig. 6 | Effect of warm intrusion on cloud condensation nuclei.** Time series of CCN number concentrations at different supersaturation (SS) during (**A**) background period and (**B**) warm intrusion period (10 min averages). **C** Violin distribution plots of hourly averaged CCN number concentration for the whole MOSAiC year data (in lila) and warm intrusion data (in pink—excluding precipitation periods). **D** Violin distribution plots of hourly averaged particle number concentration from a CPC with a lower cutoff of >2.5 nm for the whole MOSAiC year (in lila) and warm intrusion data (in pink—excluding precipitation periods). Violin plots are a combination of boxplot and a kernel distribution function on each side of the boxplots. The white circles define the median of the distribution and the edges on the inner grey boxes refer to the 25th and 75th percentiles. Both CCN and particle concentration data are cleaned from ship exhaust emissions via applying the pollution mask (see Methods section; subsection measurement of total particle concentration). CCN data include Nov. 1, 2019 to May 9, 2020 and CPC data include Nov. 1, 2019 to Sep. 30, 2020.

relatively higher sulfate contribution and oxygenated organics. During the second intrusion peak, the kappa value, drops significantly reaching 0.24 for the period with enhanced BC and Org concentrations (Supplementary Fig. 14H), consistent with the increase in the critical diameter observations (Supplementary Fig. 14C–F). Altogether, changes in aerosol size distribution, mass composition and hygroscopicity associated with the warm intrusion event affect the CCN number and properties significantly. During the intrusion period, we observed a substantial increase in CCN number and a decrease in hygroscopicity, which can result in an optically thicker cloud cover and an increase in the net surface longwave warming[5]. It is worth mentioning here that the instrumentation deployed during MOSAiC included measurements of ice nucleating particles (INPs) which can also have a significant impact on cloud properties and radiative balance. However, preliminary results show that, although INPs could be transported from mid-latitudes where they are usually more abundant, INPs did not increase in concentration during our case study[63].

To visualize the effect of the increased CCN concentration on the Arctic clouds, we include cloud observations from Polarstern (Fig. 7). Radar reflectivity measurements show deep ice-clouds at the beginning of the intrusion period, as well as periodic snowfall events before, throughout and after the intrusion period (i.e., reflectivity >0 dBZ reaching the surface). During the intrusion event, when deeper precipitation is not present, stratiform mixed-phase clouds form near the surface and up to about 2 km height. At the same time, the up and downwelling longwave radiation (Fig. 7D/E) show the increased opacity of the clouds during the intrusion period associated with the occurrence of liquid-water containing clouds or deep ice clouds. Here, the CCN increase could play an important role for the liquid-containing clouds both through direct radiative effects and by supplying this air-mass with a high CCN concentration to impact cloud processes downstream. Given the already high CCN concentration during the Arctic "background state", i.e. haze period, and the fact that the liquid-containing clouds are already optically thick (liquid water path >30 g/m², Fig. 7C–E), the impact of increased CCN on downwelling longwave radiation might not be that large[64] at the time the air-mass passed over Polarstern. Yet, the increased CCN concentrations might have larger impacts on the cloud reflection of shortwave radiation via the Twomey effect[65], which occurs over a wider range of liquid water path values. In addition to these instantaneous radiative effects, increased CCN concentrations during such warm intrusion events could affect the precipitation and lifetime of liquid containing clouds in the Arctic. A much higher CCN concentration leads to more numerous, smaller droplets, which are significantly less effective at growing to precipitation sizes. Additionally, such smaller droplets are expected to be less effective at forming ice[66]. Thus, this high concentration of CCN would inhibit precipitation from the mixed-phase clouds and therefore weaken their primary sink of moisture. Building on that point, the increase in CCN, based on our case study, could be impacting the time scales associated with the air-mass transformation. With less efficient loss of moisture and more plentiful CCN, the air-mass could sustain clouds for longer than it otherwise would have before moisture or CCN availability became a limiting factor. Overall, this potential extension of the cloud lifetime means that the cloud radiative effects would occur over a longer time and larger spatial extent during the progression of the intrusion event.

In addition, the vertical equivalent potential temperature structure of the atmosphere during the intrusion (Fig. 7B) reveals the role of

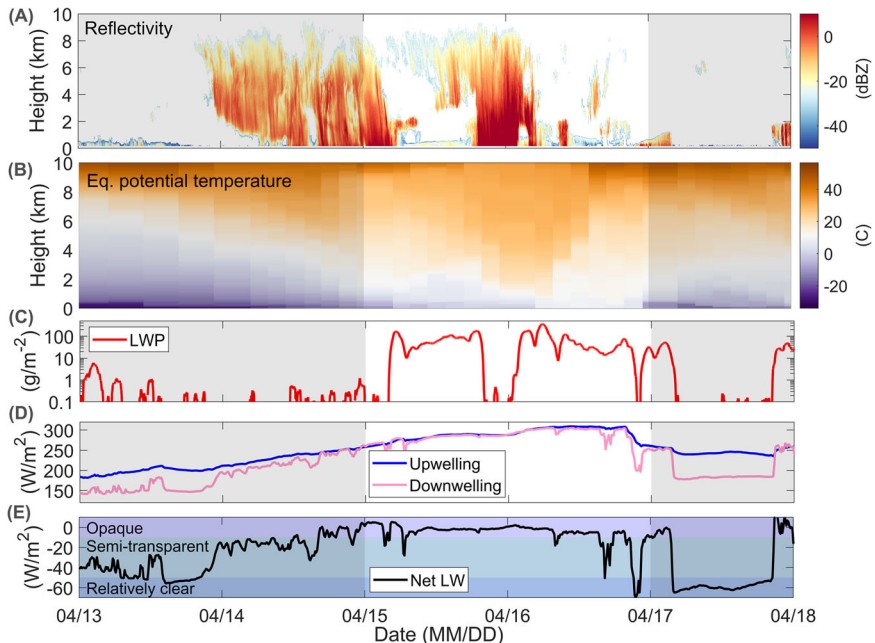

**Fig. 7 | Cloud observations during the warm intrusion event.** (A) Radar reflectivity showing the cloud base height, (B) Equivalent potential temperature showing the vertical temperature profile, (C) Liquid water path (D) Upwelling (blue) and downwelling (pink) longwave radiation and (E) the net longwave radiation. The unshaded area marks the warm air mass intrusion, the focus of this study.

vertical mixing for dispersing aerosols. Outside of the intrusion time window, the near surface was quite stratified, as is often the case in the background Arctic atmospheric state (classic stable boundary layer[11]). During the event, the near-surface both warms and becomes less stratified (i.e., weaker vertical gradient of equivalent potential temperature as a result of mixing). Part of this mixing is due to the aforementioned liquid-water clouds, which are optically thick and have strong cloud top radiative cooling, which drives buoyancy induced turbulent mixing of the atmosphere. The time evolution of the equivalent potential temperature indicates active mixing when the liquid-containing clouds are present, but not much mixing during the snowfall or background conditions. Together, the temperature gradients and FLEXPART simulations indicate high concentrations of pollutants transported at low-altitudes above the boundary layer from mid-latitudes, that are then subject to episodic vertical mixing that facilitates their transport into the boundary layer and their observation at near-surface levels. Altogether, these observations suggest that not only do aerosols impact clouds and their effects, but clouds can also impact the vertical distribution of aerosols.

## Discussion

The increased frequency of winter and springtime warm and moist airmass intrusions has gained attention in the last decade owing to their effects on the onset of summer ice-melt, clouds in the Arctic and Arctic warming[16–19]. However, none of these studies investigated the impact of such intrusions in terms of changes in aerosol microphysics and chemical composition, and resulting implications for the Arctic low concentration aerosol regime compared to air-mass source regions in lower latitudes even in the presence of Arctic haze. Our study is one of a kind, showing the short-term transformation of the Arctic environment from remote to urban-like, and the potentially impactful change for Arctic clouds and biogeochemistry both with far reaching implications on the Arctic climate, in case such intrusions become more frequent.

The Arctic wintertime background condition, known as Arctic haze is disturbed at some point in spring with the arrival of warm intrusions from lower latitudes. During the Polarstern drift, the haze condition was terminated by an extreme, record-breaking increase of near-surface temperature[10] accompanied by the arrival of an air-mass originating from Eurasia on April 15th 2020. The air-mass arriving from North-western Russia and North-eastern Europe brought higher moisture, with strong effects on Arctic cloud properties[5]. The extreme event caused an increase in aerosol number concentration and aerosol mass transforming the Arctic from a remote low concentration environment into an area comparable to an urban setting in central Europe.

Importantly, sulfate concentration reached up to $4.2\,\mu g/m^3$ (15 min average), higher than what is commonly measured in European cities[42]. This observation is remarkable and owes to two factors: (1) The source region is an important emitter of $SO_2$ and (2) the formation of $H_2SO_4$ from $SO_2$ oxidation and partitioning into the particle phase was very efficient along the transport route[67], because $SO_2$ concentrations at Polarstern were at the detection limit (1 ppb (60 second averaging time)), see also Supplementary Fig. 6.

While aerosol mass composition during background conditions was nearly invariable, the intrusion introduced highly variable composition, with elevated concentrations of $NH_4^+$ and BC. BC concentrations reached $0.5\,\mu g/m^3$ (15 min average), comparable to those observed over the Arctic Ocean near Russian sources[24] but a factor of ten higher than the annual medians observed at several Arctic land-based observatories. This can have important effects for direct aerosol-radiation interaction in the atmosphere and for surface albedo change upon deposition on the sea ice[68]. The elevated $NH_4^+$ concentration increased particle pH, allowing for a more efficient partitioning of MSA into particle phase according to our observations and a thermodynamic model. Although this single event does not allow us to draw conclusions on long term acidity of the aerosol in spring in the central Arctic, an overall decrease in aerosol acidity of the Arctic atmosphere (60–90° N) comparable to the year 1971 has been observed[69]. Therefore, an expected increase in the frequency of intrusions together with policy-imposed decreases in $SO_2$ emissions in Russia and elsewhere, might point towards a rather substantial decrease in central Arctic aerosol acidity. Decreasing acidity, in turn, might affect future partitioning of MSA and other compounds (e.g.,

organic acids) into the particle phase, and thereby influence particle hygroscopicity and hence cloud-forming potential. Our observation of decreased acidity during the intrusion is also relevant for interpretation of trends in natural aerosol tracers, such as MSA, used to investigate the changing Arctic chemical regime as a result of climate forcing, because individual intrusion events strongly impact monthly statistical values.

Besides altering the overall aerosol chemistry, the air-mass intrusion also had a direct effect on aerosol-cloud interactions. The intrusion caused the highest CCN number concentrations observed throughout the entire MOSAiC expedition year, together with the advected moisture. While the net impact of increased CCN concentrations in this environment involves a complex web of cloud microphysical processes[70], such an increase could mean that optically thicker clouds can form over the Arctic pack ice leading to stronger downwelling longwave radiation[5], which was observed to be abnormally high in this case[10], thereby positively re-enforcing the warming effect. The increase in CCN could be impacting the time scales associated with the air-mass transformation leading to a potential extension of the central Arctic clouds' lifetime leading to longer time and larger spatial extent of cloud radiative effects during such intrusion events.

As of yet, there are no such comprehensive central Arctic Ocean aerosol observations and land-based observatories only partially capture such intrusion events, highlighting the uniqueness of our measurements, and the need to more deeply understand climate and biogeochemical implications (i.e., transport of nitrogen to the central Arctic).

Together, the temperature gradients and FLEXPART simulations indicate high concentrations of pollutants transported at low-altitudes above the boundary layer from mid-latitudes, that are then subject to episodic vertical mixing that facilitates their transport into the boundary layer and their observation at near-surface levels. Altogether, these observations suggest that not only do aerosols impact clouds and their effects, but clouds can also impact the vertical distribution of aerosols. While previous studies identified the importance of studying individual mid-latitude intrusions into the Arctic, it remains unknown whether these events can directly be linked to Arctic amplification and whether their frequency is dependent on the extent of this amplification. It is however clear that pulse-injections of pollution into the Arctic in terms of aerosol number concentration, climate relevant compounds such as sulfate, organics and BC, as well as possibly environmental pollutants (which we did not measure), modify Arctic cloud microphysics and the chemical regime drastically such that these need to be taken into account in chemistry transport and climate models.

## Methods

### The expedition

The MOSAiC expedition is the most comprehensive year-round expedition into the central Arctic exploring the Arctic climate system. This study includes observations mainly in April 2020 during leg 3, as well as comparisons to measurements from the whole MOSAiC year[29]. The location of the German research vessel Polarstern during April 2020 was between 85°12'N 14°57'E and 83°55'N 17°37'E.

### Instrumentation

**The Swiss measurement container.** A full suite of state-of-the-art instrumentation (trace gases concentrations, aerosol number size distribution, aerosol mass composition, cloud condensation nuclei counter) was deployed on the bow of Polarstern (D-deck) inside the Swiss container. The temperature inside the container was maintained constant at 20 °C. Inlets were heated and RH kept below 40%. Aerosol particles and trace gases were sampled from three different inlets: (i) total inlet for sampling all particles and droplets up to 40 μm in diameter, (ii) an interstitial inlet equipped with a 1 μm cyclone for sampling particles that do not activate in cloud and fog, and (iii) a new particle formation inlet for sampling aerosol precursor gases, gas phase molecular clusters and small particles with a size of 1–40 nm[59]. A valve located inside the container switched automatically every hour between the total and interstitial inlets for an alternate sampling from both inlets. A schematic of the measurement setup and the instrumentation design and inlets used during the expedition are shown in Supplementary Fig. 15. The flows of the inlets were kept constant at 10 and 16.7 L/min for the total and interstitial inlets, respectively. The inlets protruded outside the container, had a length of 1.5 m and sampled at ~15 m above sea level.

**Measurement of aerosol chemical composition.** The aerosol mass chemical composition was measured using an Aerodyne High resolution Time-of-Flight Aerosol Mass Spectrometer, HR-AMS[71]. The AMS was equipped with a 1 μm aerodynamic lens allowing the detection of non-refractory (<600 °C) components of particulate matter with a diameter smaller than 1 μm (PM$_1$). The AMS was located in the Swiss container, and sampled alternatively every hour from the total and interstitial inlets. The AMS was operated in two modes simultaneously: the 'mass spectrum (MS) mode' and the "particle time-of-flight (PTOF) mode" and the time resolution is 90 s.

Several times per month zero measurements were conducted by installing a high-efficiency particulate absorbing (HEPA) filter in front of the inlet, and were used for background correction. To remove any additional gas phase interference of chloride, the signal during which the AMS was sampling through a HEPA filter was subtracted from the total chloride signal. Throughout the expedition, on-site calibrations for ionization efficiency of NO$_3^-$, NH$_4^+$ and SO$_4^{2-}$ were conducted using monodisperse number concentration-defined NH$_4$NO$_3$ and (NH$_4$)$_2$SO$_4$ particles. During April 2020, which is the period of focus in this study, the detection limits for sulfate, nitrate, ammonium, chloride, and organics were 0.048, 0.026, 0.0065, 0.034, and 0.19 μg m$^{-3}$, respectively, calculated from 3 times the standard deviations of the filter periods, and valid for a time resolution of 90 s. The measured concentrations of nitrate (NO$_3^-$) and chloride (Cl$^-$) were insignificant, also during the intrusion period, did not show any variation, and were close to the detection limit, thus were excluded from further analysis.

**Determination of particulate phase methanesulfonicacid (MSA).** MSA is calculated following the method and Eq. (1) below[72]:

$$m_{MSA} = \frac{m_{CH_2SO_2} + m_{CH_3SO_2} + m_{CH_4SO_3}}{0.147} \tag{1}$$

**Development of pollution mask for AMS data.** In order to eliminate periods contaminated with Polarstern emissions, we developed a pollution mask for the AMS data. The method is based on finding the similarity between the mass spectrum at each data point and a ship pollution spectrum. The ship pollution spectrum has been derived from the average of three distinct spectra from contaminated periods in the beginning, middle and end of April. The selection of the spectra is based on the observation of contamination in other instruments, the wind direction arriving from the ship stack, as well as a noticeable increase in ship emission markers in the AMS time series, namely C$_4$H$_7$, C$_4$H$_9$, C$_5$H$_7$, C$_5$H$_9$, C$_6$H$_7$, C$_6$H$_9$, C$_7$H$_7$, C$_7$H$_9$, C$_8$H$_7$, and C$_8$H$_9$[73]. The similarity between each of the data points and the ship spectra is determined using the cosine similarity method. The similarity is measured by the cosine of the angle between two vectors using the

following Eq. (2):[74]

$$\text{similarity} = \cos\theta = \frac{A \cdot B}{||A|| \cdot ||B||} = \frac{\sum\limits_{i=1}^{n} A_i B_i}{\sqrt{\sum\limits_{i=1}^{n} A_i^2} \cdot \sqrt{\sum\limits_{i=1}^{n} B_i^2}} \quad (2)$$

where A and B are the spectra at each point and the polluted spectra, respectively and $A_i$ and $B_i$ are the components of vectors A and B, respectively. The larger $\cos\theta$, the more similar are the point spectrum and the exhaust spectrum. In Supplementary Fig. 16 we show the histogram of all the data and those affected by the ship exhaust owing to a wind direction from between 120 and 240° from the ship stack. We chose a $\cos\theta$ of 0.56 as the threshold above which data are considered contaminated by the ship exhaust. This threshold is chosen as it corresponds to 80% of the data points during which the wind direction was outside the 120 and 240° (dashed line in Supplementary Fig. 16). This threshold retains 47.7% on the data points during April, as clean. Such a pollution flag can be used for multiple purposes besides identifying contaminated periods and thereby the threshold can be modified according to the purpose of the implementation.

**Measurement of black carbon (BC).** An aethalometer (AE33, Magee Scientific, Berkeley, USA) was used to continuously measure the mass concentration of equivalent black carbon (eBC) at 7 defined wavelengths in a time resolution of one second. The instrument was placed in the Swiss Container behind the switching valve. The inlet flow of 2 liters per minute (lpm) was checked biweekly. The BC measured at 880 nm was used in this work.

**Measurement of particle number size distribution.** The particle number size distribution was measured using a Scanning Mobility Particle Sizer (SMPS[75]) within the Aerosol Observing System (AOS) container operated by the United States Department of Energy Atmospheric Radiation Measurement (ARM) user facility[76,77]. The AOS container, located on the bow of Polarstern, contained the SMPS, which shared a total aerosol inlet, 5 m in length, extending from the top of the AOS container at a height of approximately 18 m above sea level. The AOS container was located adjacent to the Swiss container. For the detection and exclusion of periods contaminated by ship emissions, an automated detection algorithm was applied based on the method developed by Beck et al.[78].

**Measurement of total particle concentration.** The total particle number concentration was measured using condensation particle counter (CPC-model 3025 by TSI) located inside the Swiss Container. The CPC measures the total particle number concentration of particles with diameters of 3 nm (50% counting efficiency) and larger. The data were collected at 10 s intervals. The instrument was connected to the interstitial inlet (Supplementary Fig. 15). The sample flow of the CPC was set to 0.3 L/min during the entire expedition and was checked daily. Weekly zero tests with HEPA filters were performed. For the detection and exclusion of periods contaminated by ship emissions, an automated detection algorithm was applied[78].

**Measurement of gaseous methanesulfonicacid.** The MSA concentrations were measured using a Chemical ionization Atmospheric Pressure interface Time-of-Flight spectrometer (CI-APi-ToF) using $NO_3^-$ as a reagent ion[79,80]. The data were analyzed using a tofTools package based on MATLAB software[81]. The CI-APi-ToF instrument was calibrated after the campaign using the method presented in previous studies[82] to ensure quality of the measurement resulting in a calibration factor of $6 \times 10^9$ molecules per ion count while accounting for line losses.

**Measurement of cloud condensation nuclei.** Polydisperse CCN measurements were performed using a commercial DMT CCN counter scanning at five different SSs (0.1%, 0.2%, 0.3%, 0.5%, and 1.0%) estimated to be 0.18, 0.21, 0.29, 0.53, 0.78 % after instrument calibration[83]. The critical activation diameter was calculated by integrating the particle number size distribution from the largest diameter to that diameter at which the integrated particle number equaled the measured CCN number concentration[39]. CCN data were collected between Oct 15th 2019 and May 9th 2020.

**Measurement of $H_2O$ vapor.** Water vapor was measured by cavity ring-down spectroscopy (CRDS) using a commercial Picarro instrument (model G2401) connected to the interstitial inlet (see Supplementary Fig. 15).

**Meteorological variables.** We used meteorological data (temperature, relative humidity, wind direction) from the weather stations at 29 m above sea-level located on board Polarstern[84]. Surface broadband, upwelling and downwelling longwave radiation were measured on the sea ice adjacent to Polarstern using a pair of pyrgeometers operated by the ARM program[85].

**Cloud and precipitation observations.** The vertical profile of clouds and precipitation was observed using the Ka-band ARM Zenith Cloud Radar (KAZR[86]) operated by the ARM Program from a container adjacent to the AOS. In particular the radar reflectivity, mean Doppler velocity, and Doppler spectrum width provide information on the cloud and precipitation processes and type, as well as some insight into atmospheric mixing associated with cloud processes. The cloud liquid water path (LWP) is obtained from the ShupeTurner cloud microphysics product[87], based on microwave radiometer measurements from Polarstern's upper deck. Precipitation mass was observed by the ARM Program using a laser disdrometer (LDIS), which measures the drop size spectra and fall velocity of hydrometeors during precipitation events[88].

**Boundary layer height.** Boundary layer height (BLH) was calculated using meteorological profiles from radiosondes, which were launched at least four times per day from the stern deck of the Polarstern throughout the entire MOSAiC year[89] and the DataHawk2 (DH2) uncrewed aircraft system[90], which was flown during legs 3 and 4 of MOSAiC (March – July 2020) whenever weather and airspace criteria were met[91]. Between 00:00 on 13 April and 00:00 on 18 April, 2020, 32 radiosondes were launched and 8 DH2 flights were conducted. During events of interest, such as the warm air intrusion event discussed in this paper, the frequency of radiosonde launches was increased, resulting in a total of 14 radiosonde profiles for the 15th–16th of April, along with one DH2 flight. To determine BLH, a bulk Richardson-based approach was taken in which the BLH is identified at the altitude after which the bulk Richardson number consistently exceeds a critical value of 0.5. This BLH-detection method has been determined to be the most successful among other options at accurately identifying BLH in the central Arctic according to Jozef et al.[92].

**FLEXPART simulations**

To determine the origin of the observed air masses and the contribution from pollution sources, we performed backward simulations for a passive air tracer without removal processes and a black carbon (BC) tracer with wet and dry removal with the Lagrangian particle dispersion model FLEXPART v10.4[31]. The simulations were based on hourly meteorological data from the ERA5 reanalysis with 0.5° × 0.5° resolution. Every three hours a cluster of 100,000 atmospheric particles was initialized at Polarstern's location and traced backward for 7 days. An archive of the model results can be found at https://img.univie.ac.at/webdata/mosaic. The FLEXPART output consists of

3-dimensional fields of emission sensitivity sometimes also referred to as "source-receptor relationship"[93]. It represents the influence a unit emission flux would have on the passive tracer concentration at the ship location. Of particular interest is the emission sensitivity close to the surface (below 100 m, the lowest model output layer), since most emissions occur at the surface. When multiplying the emission sensitivities with emission data, we also obtain source contribution maps, and by spatial integration we obtain a concentration at the ship location. To describe the source contribution of BC and $SO_4^{2-}$ from anthropogenic activities, we coupled the footprint emissions sensitivity with emission inventories from the ECLIPSE database (Evaluating the Climate and Air Quality Impacts of Short-Lived Pollutants: https://iiasa.ac.at/web/home/research/researchPrograms/air/ECLIPSEv5a.html). The temperature and specific humidity along the trajectories were extracted from ERA5 and averaged for all the air masses below 1500 m.

### Determination of aerosol hygroscopicity

The hygroscopicity parameter, kappa ($\kappa$) was derived using the volume weighted aerosol composition measured using the AMS data, Eq. (3):

$$\kappa = \sum_i \varepsilon_i \, \kappa_i \tag{3}$$

where $\varepsilon_i$ and $\kappa_i$ are the volume fraction and the hygroscopicity parameter and of the individual components[94].

### Determination of aerosol acidity

**Ammonium Balance.** The degree of aerosol neutralization ($NH_4^+$ Balance, Eq. (4)) is given by the ratio of the measured ammonium concentration ($NH_4^+$ measured) to the amount of $NH_4^+$ needed to neutralize $SO_4^{2-}$, $NO_3^-$ and $Chl^-$ ($NH_4^+$ predicted, Eq. (5))[57,95].

$$NH_4^+{}_{Balance} = \frac{NH_4^+{}_{measured}}{NH_4^+{}_{predicted}} \tag{4}$$

$$NH_4^+{}_{predicted} = M_{NH_4^+} \times \left( 2 \times \frac{SO_4^{2-}}{M_{SO_4^{2-}}} + \frac{NO_3^-}{M_{NO_3^-}} + \frac{Chl^-}{M_{Chl^-}} \right) \tag{5}$$

**Extended aerosol inorganic model (E-AIM).** E-AIM was used to model the pH of the aerosol using the measured aerosol mass composition (http://www.aim.env.uea.ac.uk/aim/aim.php, last access: 21.10.2021). The model was further used to calculate the partitioning of MSA between gas and particle phase[96]. The MSA properties are based on the values reported in Baccarini et al. [60]. While there is a difference in the absolute value between measured and predicted partitioning, the relative difference between the background, first and second peaks remains visible. The deviation from the absolute values could be attributed to the uncertainties associated with the MSA thermodynamic properties or the equilibrium state of the gases and particles, which might have not been reached given the abrupt change in temperature.

## Data availability

Particle number size distribution, radiation, cloud and precipitation data were obtained from the Atmospheric Radiation Measurement (ARM) User Facility, a U.S. Department of Energy (DOE) Office of Science User Facility managed by the Biological and Environmental Research Program. The ARM datasets are available via the ARM Data Discovery tool: https://adc.arm.gov/discovery/#/.

Particle number concentration measured in the Swiss aerosol container during MOSAiC 2019/2020 is open access available at https://doi.org/10.1594/PANGAEA.941886. Aerosol mass composition and cloud condensation nuclei data from the Swiss container can be provided by request from the corresponding authors until December 31, 2022. After that date, the data will be publicly accessible via the online repository PANGAEA as per the MOSAiC data policy.

An archive of the FLEXPART model outputs for the whole campaign are available at https://img.univie.ac.at/webdata/mosaic.

## Code availability

FLEXPART's source code and documentation are freely available at https://www.flexpart.eu/.

Analysis Software Resources for Aerosol Mass Spectrometric data are openly available at https://cires1.colorado.edu/jimenez-group/wiki/index.php/ToF-AMS_Analysis_Software.

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

## Acknowledgements

Data reported in this study were produced as part of the international Multidisciplinary drifting Observatory for the Study of Arctic Climate (MOSAiC) expedition with the tag MOSAiC20192020, with activities supported by Polarstern expedition AWI_PS122_00. We also thank all those who contributed to MOSAiC[97].The authors would like to thank the teams at the Paul Scherrer Institute and the University of Helsinki for their land-based support during the MOSAiC expedition. J.S. holds the Ingvar Kamprad chair for extreme environments research, sponsored by Ferring Pharmaceuticals. The authors thank Jakob Pernov, Roman Pohorsky and Benjamin Heutte for fruitful discussions. Funding: This research was funded by the Swiss National Science Foundation (grant 200021_188478) and the Swiss Polar Institute (J.S.). UAS observations and data processing and analysis was supported by the US National Science Foundation (OPP 1805569) and the NOAA Physical Sciences Laboratory (G.B.). European Research Council ERC (GASPARCON—grant no 714621) is acknowledged (L.Q.). This work was supported by the Academy of Finland (project 334514) and the EMME-CARE project which received funding from the European Union's Horizon 2020 Research and Innovation Programme, under grant agreement no. 856612 (T.J.). This

work was 994 supported by Academy of Finland via project (333397) and Atmosphere and Climate Competence Center (337549) and University of Helsinki ACTRIS-HY (T.P.). The US Department of Energy Atmospheric System Research Program (DE-SC0019251, DE-SC0021341) is acknowledged for financial support (M.D.S.). Support by the Swiss National Science Foundation Ambizione grant PZPGP2_201992 is acknowledged (K.R.D.).

## Author contributions

Conceptualization of the study: L.D., J.S. Measurements: J.S., G.d.B., I.B., G.J., L.Q., T.J., M.B., Z.B., M.D.S. Data analyses: L.D., I.B., A.B., H.A., L.Q., J.S., G.d.B., G.J., S.H., S.B., T.J., M.B., M.D., K.R.D., M.D.S. Results interpretation: L.D., J.S., G.d.B., A.S., T.P., T.J., K.R.D., MDS Supervision: J.S. Writing—original draft: L.D. Writing—Materials and Methods: L.D., H.A., G.J., A.S., S.B., M.D.S. Commenting: all.

## Competing interests

The authors declare no competing interests.
