## [Peer Review File · Nature Communications]

Review

The study titled “**A Central Arctic Extreme Aerosol Event Triggered by a Warm Air-Mass Intrusion**” by Dada et al. investigates the aerosol concentration and composition during a warm air-mass intrusion into the Arctic as measured during the MOSAiC expedition. The measurements represent a one-of-a-kind dataset from the remote central Arctic Ocean, a region difficult to access and hence of sparse data coverage. The authors show that the number concentration of CCN substantially increases during the air-mass intrusion, which, apart from changes in temperature and humidity during the intrusion, can impact local cloud cover and transform the Arctic from a pristine into an urban-like environment.

The manuscript is well-written and represents novel measurements from the remote Arctic Ocean. The findings highlight the strong impact of air-mass intrusions on the Arctic environment. Even though not very surprising, this dataset and the subsequent implications show for the first time how pollution can be effectively introduced into the Arctic environment during intrusion events. Hence, I encourage publication in Nature Communications, however, I have a couple of comments which should be considered prior to final publication.

General comments

1. The authors indicate the relevance of the measured elevated aerosol concentration for the ambient CCN concentration, however, aerosols containing e.g., organic materials could also act as ice nucleating particles (INPs; Kanji et al., 2017). However, any increase in INP concentration has not been discussed at all in the manuscript. If the authors also measured INP concentrations, could they include any data for this as well? As INP measurements from the Arctic are also particularly sparse, any new insights in this respect would be very valuable. In addition, increased INP concentrations have been shown to potentially glaciate the clouds (e.g., Stevens et al., 2019) and could thus have an opposing effect to the increased CCN concentration during the air-mass intrusion. If no measurements of INP are available, including any potential impact in the discussion would be needed, as currently it reads as if aerosols are solely important for CCN.
2. The authors speculate about to which extent the increased CCN concentrations could impact Arctic clouds and their properties. Given the huge number of observations collected during the MOSAiC expedition, I am wondering if the authors could show some cloud observations from either the *Polarstern* vessel or even from satellite data, to get an idea of the impact of the additional aerosols? I think the study would benefit a lot from putting the observations into context of the actual impact. I also would like to point out that the background CCN concentration of 150 cm⁻³ as a result of the Arctic haze is not particularly low, hence, additional aerosols might have less of an impact as compared to a CCN-limited environment as for example can be observed in the Arctic in summer.
3. I find some of the figures very hard to read. I understand that the authors want to use the same projection for each figure, however, if the observed signals are locally constrained (e.g., Figures 1d, S5, and S6) it is very difficult to detect the signal, especially the actual values. In Figure S6 the authors zoom in into the Kola Peninsula, which for example could be one option of emphasizing small-scale signals.

In the same figures it would also be helpful to remove the very small titles on top of each panel and label them as ‘Background’, ‘First peak’, and ‘Second peak’.

Specific comments

Abstract

Line 35: The potential effects on biogeochemistry are also mentioned in the 'Implications' section, but are not further discussed or explained. I would remove this here or replace it by 'environmental properties' (which would also include the effect on surface albedo, as briefly discussed in the manuscript) or something similar.

I know that the authors are limited in length, but in my opinion, it would be worth mentioning in the Abstract that the presented data are unique measurements from the remote central Arctic Ocean, given that aerosol measurements from the high Arctic are still sparse.

Introduction

Line 51: It might be worth adding "nor quantified" as to my knowledge the measurements presented here are quite unique.

Lines 63-64: This statement is a little confusing, doesn't the atmospheric circulation affect the synoptic weather conditions, not the other way around?

Same lines and following sentence: I would also argue that the chemical composition and properties also depend on the (local) emission sources.

Line 67: To my understanding, Arctic haze is most prominent in spring (this is also stated in Freud et al., 2017 and others).

Line 87: Related to my comment 2, in order to make this statement I think it would be worth showing some cloud property observations in the manuscript.

Results

Lines 95-96/Figure S1: would it be possible to also show the intrusion as a vertical profile to see the vertical extent of the anomalies? Or was this temperature increase only seen at the surface?

Lines 97-103: This reads more like an introduction and is a repetition, as air-mass intrusions have already been introduced in the introduction before (line 52ff), hence, I would suggest to remove this paragraph here.

Figure S2: Does the figure show surface (as indicated in the figure caption) or 2m (as indicated on top of the figure) temperature? Also, zooming in a bit into the Arctic (but still capturing the extent of the intrusion) would increase the visibility of the marked locations.

Line 108: Related to my previous comment, here it would be nice to have a vertical profile of temperature during the intrusion.

Figure 1D: What does the number on the lower right corner indicate?

Line 136: here the authors mention that the background aerosol concentration is representative of haze conditions, while in the introduction only winter is mentioned as the main period for Arctic haze. Please adjust this in the introduction.

Line 160: the statement that the aerosol has been transformed during long-range transport is only discussed in the next section, hence, I would not make this statement here yet.

Line 164: From the relative contributions reported in lines 173-174 and from Figures 3 and S7, the mass composition of the aerosol during the background period and the intrusion doesn't seem so different, rather the absolute mass/concentration is different. I would suggest slightly rewriting this paragraph, so this difference becomes clear.

Same line: Why is not Figure 3 referenced here? I also don't really understand the added value of showing Figure S7 in addition to Figure 3, which is already very comprehensible. Given the large number of supplementary figures, I would consider removing Figure S7.

Lines 190-195: I don't really understand why these peaks would not have been observed at other Arctic observatories, but on the *Polarstern*. Could the authors elaborate on the reason for this? Currently, I find this paragraph a little confusing.

Line 223: Are these correlations significant, i.e., did the authors perform a significance test?

Figure S10, caption (line 1058): again, did the authors perform a significance test? If not, I would rephrase "significant difference".

Figure 5: It might be helpful to rename "all data" to "whole expedition" or similar, as in the previous figures "all data" referred to April 2020 and not the whole year.

Line 390: increased CCN concentrations do not necessarily lead to an increase in net surface longwave warming, in fact, Eirund et al. (2019) showed that low-level clouds over sea ice perturbed by CCN become more optically thick, however, the effect in net surface LW remains small. There, the change in cloud structure might be responsible for the radiative response of the clouds to CCN concentrations changes, as stratocumulus clouds over the ocean show a larger response in net surface LW. Note that in Eirund et al., 2020 no CCN perturbations were applied, so this reference might be a bit confusing here. The radiative response in the low-level clouds in this study was solely driven by the temperature and moisture perturbations.

Implication

Line 406: I would rather show evidence in form of cloud observations here or (if available) cite publications related to this event.

Line 428: why extreme conditions?

Line 440: consider citing some other studies here which have looked at the CCN impact in Arctic clouds

Line 444: "biogeochemical implications" has been mentioned before, but what specifically is meant here? Ocean biogeochemical changes?

Line 452: "short and long-time scales" is there any evidence of a long-term effect of aerosol injections on Arctic cloud microphysics from either the available measurements or previous studies?

Methods

Line 515: Was MOSAiC really the first year-round expedition in the Arctic? To my knowledge, SHEBA was the first (and until MOSAiC the only) one (e.g., Shupe et al., 2006)

Line 526: Do the authors mean “heated”?

References

Eirund et al., 2019: Response of Arctic mixed-phase clouds to aerosol perturbations under different surface forcing, *Atmos. Chem. Phys.*, 19, <https://doi.org/10.5194/acp-19-9847-2019>

Kanji et al., 2017: Overview of Ice Nucleating Particles, *Met. Monographs*, 58, <https://doi.org/10.1175/AMSMONOGRAPHS-D-16-0006.1>

Shupe et al., 2006: Arctic Mixed-Phase Cloud Properties Derived from Surface-Based Sensors at SHEBA, *Journal of Atmospheric Sciences*, 63, <https://doi.org/10.1175/JAS3659.1>

Stevens et al., 2019: A Model Intercomparison of CCN-Limited Tenuous Clouds in the High Arctic, *Atmos. Chem. Phys.*, 18, <https://doi.org/10.5194/acp-18-11041-2018>

Review for „A Central Arctic Extreme Aerosol Event Triggered by a Warm Air-Mass Intrusion“ by Dada et al. submitted to Nature Communications

Summary & Recommendation

This study provides an in-depth characterization of aerosol composition and air mass history during the strongest warm air mass intrusion observed during the MOSAiC field campaign in the high Arctic. The intrusion was characterized by anomalously warm and moist near-surface air and significantly higher aerosol concentrations for a duration of about two days (15.-16.04.2020). Furthermore, these particles were likely considerably „younger“ (<2 days) than average spring time Arctic haze particles observed during the reference period (02.-03.04.). Furthermore, different aerosol compositions and size distributions were observed and aerosols were characterised by different hygroscopicity and acidity. All of these changes may impact aerosol-cloud -, and aerosol-radiation interactions, as well as atmospheric chemistry and secondary organic nucleation rates. Currently our understanding of the aerosol impact of air mass intrusions into the high Arctic, their impact on the local biogeochemistry and aggregated importance (i.e. across several events) under climate change and Arctic amplification remains unclear.

Dada et al. show an in-depth analysis of such an event which was missed by ground stations of similar latitude using a wide array of instruments available during the MOSAiC campaign. This in itself highlights the issues with sampling these events in a region where ground-based observations are sparse.

The study is novel, well-conceived, well written, and very carefully analysed. It is the first analysis of this kind presented in the scientific literature and I recommend publication in Nature Communications once the minor points mainly regarding clarifications are addressed.

Minor Revisions

L36: “[...],but are expected to intensify” I personally find this statement too strong. While the changes in aerosol concentrations and their impact on chemistry, clouds and radiation within the warm air intrusion is undoubtedly significant, these events (as you also state) are short-lived. Prolonged impacts or aggregate impacts across multiple intrusions on long time scales are not yet known. It could be thus be that while these events occur more often, the aerosol-driven effects associated with warm-air intrusions are insignificant on longer time scales.

L39-L87: The introduction is very well written and comprehensive. My only suggestion would be to consider to incorporate 1-2 sentences on alternate approaches in characterizing intrusions in terms of their aerosol change in terms of AOD using remote sensing. E.g.: <https://acp.copernicus.org/preprints/acp-2021-805/> and references therein.

Fig 1: Generally it is not clear to me what it means to plot a variable “along the emission sensitivity”. Emission sensitivity is defined in L653ff as the impact of a unit of emission on the tracer concentration at the ship. So are you looking along a gradient of emission sensitivities and what is the range of this gradient: 0-1? Generally do you consider all air masses with an emission sensitivity larger than 0, or do you use another threshold? What does it mean to show temperature at 1500m along the emission sensitivity? In the caption you write that its for emission sensitivities below 1500m. Is that in the same column? Or do you mean the air mass had previously been near the surface with a layer of an emission sensitivity>0? For me some more detail in the text is needed here to really understand how you use this concept.

Fig1c: Why do you show these maps at 1500m altitude? This is above the warm air intrusion as you state in line 108 and considerably above the BL, which the emissions need to be mixed into.

Generally it would be nice to see a vertical cross section along the warm air intrusion in addition to the horizontal slices. This would shed light on its vertical structure and help understanding how exactly the polluted air mass is in contact with the BL. This doesn't have to be along the emission sensitivity filtered fields, but could simply show the thermodynamic structure as its shown in Fig S2 in terms of its horizontal extend.

Fig1: 3rd columns. Rows C and E show these ripple structures along the temperature and altitude. Can you explain these?

Fig1 caption: "Altitude profile of the emission sensitivities shown in panel (C)" Revise wording. Panel C does not show emission sensitivity, but temperature fields along the emission sensitivity.

L240ff: During the first peak the correlation is similarly low. Can you explain this? I assume its due to the different source regions and resulting differences in composition?

L419ff: You state that aerosol-radiation interactions may be important. Could you not diagnose these during the event using ground-based measurements on radiation obtained during MOSAiC?

Fig2: Do you have a hypothesis for why the accumulation mode leads the Aitken mode during the second peak?

L661: ECLIPSE: please write out acronym

Edits

L40: Arctic amplification in my opinion is a phenomenon, not a process.

L50: "and the related" -> "and related"

L106: double spacing

L702: missing space

Reviewer #1

The study titled “**A Central Arctic Extreme Aerosol Event Triggered by a Warm Air-Mass Intrusion**” by Dada et al. investigates the aerosol concentration and composition during a warm air-mass intrusion into the Arctic as measured during the MOSAiC expedition. The measurements represent a one-of-a-kind dataset from the remote central Arctic Ocean, a region difficult to access and hence of sparse data coverage. The authors show that the number concentration of CCN substantially increases during the air-mass intrusion, which, apart from changes in temperature and humidity during the intrusion, can impact local cloud cover and transform the Arctic from a pristine into an urban-like environment.

The manuscript is well-written and represents novel measurements from the remote Arctic Ocean. The findings highlight the strong impact of air-mass intrusions on the Arctic environment. Even though not very surprising, this dataset and the subsequent implications show for the first time how pollution can be effectively introduced into the Arctic environment during intrusion events. Hence, I encourage publication in Nature Communications, however, I have a couple of comments which should be considered prior to final publication.

We thank the reviewer for their thoughtful comments that help improve our manuscript. We revised the manuscript following the reviewers’ suggestions and responded to the point-by-point comments in blue. Additions to the text are shown in *italics*.

General comments

1. The authors indicate the relevance of the measured elevated aerosol concentration for the ambient CCN concentration, however, aerosols containing e.g., organic materials could also act as ice nucleating particles (INPs; Kanji et al., 2017). However, any increase in INP concentration has not been discussed at all in the manuscript. If the authors also measured INP concentrations, could they include any data for this as well? As INP measurements from the Arctic are also particularly sparse, any new insights in this respect would be very valuable. In addition, increased INP concentrations have been shown to potentially glaciate the clouds (e.g., Stevens et al., 2019) and could thus have an opposing effect to the increased CCN concentration during the air-mass intrusion. If no measurements of INP are available, including any potential impact in the discussion would be needed, as currently it reads as if aerosols are solely important for CCN.

We agree with the reviewer on the importance of including INP measurements and discussions related to warm air mass intrusions in the Arctic. Actually, there were measurements of INP on the ship, and consisted of a size-resolved Davis Rotating-drum Unit for Monitoring (DRUM) for immersion INP measurements and disposable sterile filter unit samplers for immersion INP measurements and DNA sequencing specific for characterizing the airborne microbial community. Detailed information about the measurements and a general overview of the results can be found in Creamean et al. (2021). An additional dedicated manuscript is currently under review with Nature Communications (Creamean et al. 2022). Based on the measured data, the INP concentration does not increase with the arrival of the warm air intrusion. Although INPs are expected to be transported from mid-latitudes where they are usually more abundant than in the Arctic, INPs did not increase in concentration during our case study. Following the reviewer’s suggestion, and to avoid any misunderstanding related to the role of INPs during this specific intrusion, we added the following information to the results section:

It is worth mentioning here that the instrumentation deployed during MOSAiC included measurements of ice nucleating particles (INPs) which can also have a significant impact on cloud properties and radiative balance. Preliminary results show that, although INPs could be transported from mid-

latitudes where they are usually more abundant, INPs did not increase in concentration during our case study (Creamean et al., 2021).

2. The authors speculate about to which extent the increased CCN concentrations could impact Arctic clouds and their properties. Given the huge number of observations collected during the MOSAiC expedition, I am wondering if the authors could show some cloud observations from either the *Polarstern* vessel or even from satellite data, to get an idea of the impact of the additional aerosols? I think the study would benefit a lot from putting the observations into context of the actual impact. I also would like to point out that the background CCN concentration of 150 cm^{-3} as a result of the Arctic haze is not particularly low, hence, additional aerosols might have less of an impact as compared to a CCN-limited environment as for example can be observed in the Arctic in summer.

As per the reviewers' recommendation, we added cloud observations from *Polarstern* to visualize the impact of the intrusion on the central Arctic atmosphere (Fig. 6 in the main text). Here, we included the radar reflectivity to show the cloud height and the clouds' liquid water path (LWP). During the air-mass intrusion, we observe mixed-phase clouds that form near the surface and up around 2 km height, which could be impacted by CCN concentration. During the start of the intrusion period, deep ice-clouds, which precipitate, are observed. Additionally, the intrusion period is interrupted by heavy snow fall. Both of these precipitating periods inhibit the formation of liquid clouds at lower levels (seeder-feeder mechanism). When significant deep precipitation is not occurring, classic Arctic mixed-phase stratiform clouds are present. The observed CCN increase with the warm air advection could play an important role for those clouds and for providing a source of new aerosols/CCN for the following "background state" of the Arctic (i.e., the increased CCN concentration will remain with this advected air mass for an extended time downstream). We also included the up and downwelling longwave (LW) radiation (Fig. 6d) as well as the net longwave radiation (Fig. 6e). The net LW shows the increased opacity of the clouds during the intrusion period associated with lots of cloud particles and liquid-water containing clouds.

As the reviewer pointed out, the impact of increasing CCN on downwelling longwave radiation might not be that large in this case because the background concentration is already pretty high given that the clouds are already optically thick ($\text{LWP} > 30 \text{ g/m}^2$). In this case, there is little change in cloud emissivity because the cloud is already effectively a blackbody emitter. There might be larger impacts on the shortwave radiation, which reaches its "saturation" point at a higher cloud LWP. Also, in addition to these radiative effects, there are other potential implications of increased CCN during these events:

- (1) Precipitation: for the liquid containing clouds, a much higher CCN concentration leads to more numerous, smaller droplets. These are significantly less effective at growing to precipitation sizes. Additionally, these smaller droplets are expected to be less effective at forming ice (Lance et al., 2004; Norgren et al., 2018). Thus, this higher concentration of CCN would mean less precipitation from the mixed-phase clouds and therefore a weakened precipitation sink of the atmospheric moisture.
- (2) Lifetime: Building on that point, the increase in CCN could impact the time scales associated with the air-mass transformation. With less efficient loss of moisture, the clouds would last longer than they otherwise would have. That lifetime is related to both the total moisture budget and the CCN budget. So presumably the clouds would last much longer before they become limited by moisture or particles. As a result of this increased cloud lifetime, the cloud radiative effects would also last longer and thus have an impact over a longer time and larger spatial scale. The discussion we present here is based on a single case study only and more investigation is needed.

By including, in addition, the vertical temperature structure of the atmosphere during the intrusion, another characteristic of the intrusion is revealed: vertical mixing. We find that outside of the intrusion time window, the near surface was quite stratified, as is often the case in the background Arctic atmospheric state (classic Arctic stable boundary layer). During the event, the near-surface both warms

and becomes less stratified (i.e., more mixing). Part of this mixing is due to the aforementioned liquid-water clouds, which are optically thick, and have strong radiative cooling to space, driving turbulent mixing of the atmosphere. The temporal transition in equivalent potential temperature vertical gradients (Fig. 6b) shows how active mixing occurred during the intrusion periods, compared to not much mixing during the strong snowfall or background conditions. Overall, the temperature gradients and FLEXPART simulations indicate high concentrations of pollutants transported at low-altitudes above the boundary layer from mid-latitudes, that are then subject to episodic vertical mixing that facilitates their transport into the boundary layer and their observation at near-surface levels. The liquid water clouds present during the intrusion period, and specifically their cloud top radiative cooling that drives buoyant circulations over some depth, likely played a major role in mixing the aerosol down towards the surface.

We added the following figure and associated discussion to the ‘Implications’ section:

Fig. 6. Cloud observations during the warm intrusion event. a) Radar reflectivity showing the cloud base height, b) Equivalent potential temperature showing the vertical temperature profile, c) Liquid water path, d) Upwelling (blue) and downwelling (red) longwave radiation, and e) the net longwave radiation. The unshaded area marks the warm air mass intrusion, the focus of this study.

To visualize the effect of the increased CCN concentration on the Arctic clouds, we include cloud observations from Polarstern (Fig. 6). Radar reflectivity measurements show deep ice-clouds at the beginning of the intrusion period, as well as periodic snowfall events before, throughout and after the intrusion period (i.e., reflectivity > 0 dBZ reaching the surface). During the intrusion event, when deeper precipitation is not present, stratiform mixed-phase clouds form near the surface and up to about 2 km height. At the same time, the up and downwelling longwave radiation (Fig. 6d) show the

increased opacity of the clouds during the intrusion period associated with the occurrence of liquid-water containing clouds or deep ice clouds. Here, the CCN increase could play an important role for the liquid-containing clouds both through direct radiative effects and by supplying this air-mass with a high CCN concentration to impact cloud processes downstream. Given the already high CCN concentration during the Arctic 'background state', i.e. haze period, and the fact that the liquid-containing clouds are already optically thick (liquid water path $> 30 \text{ g/m}^2$, panels c-e), the impact of increased CCN on downwelling longwave radiation might not be that large (Eirund et al., 2019) at the time the air-mass passed over Polarstern. Yet, the increased CCN concentrations might have larger impacts on the cloud reflection of shortwave radiation via the Twomey effect (Twomey, 1974), which occurs over a wider range of liquid water path values. In addition to these instantaneous radiative effects, increased CCN concentrations during such warm intrusion events could affect the precipitation and lifetime of liquid containing clouds in the Arctic. A much higher CCN concentration leads to more numerous, smaller droplets, which are significantly less effective at growing to precipitation sizes. Additionally, such smaller droplets are expected to be less effective at forming ice (Lance et al., 2004; Norgren et al., 2018). Thus, this high concentration of CCN would inhibit precipitation from the mixed-phase clouds and therefore weaken their primary sink of moisture. Building on that point, the increase in CCN, based on our case study, could be impacting the time scales associated with the air-mass transformation. With less efficient loss of moisture and more plentiful CCN, the air-mass could sustain clouds for longer than it otherwise would have before moisture or CCN availability became a limiting factor. Overall, this potential extension of the cloud lifetime means that the cloud radiative effects would occur over a longer time and larger spatial extent during the progression of the intrusion event.

In addition, the vertical equivalent potential temperature structure of the atmosphere during the intrusion (Fig. 6b) reveals the role of vertical mixing for dispersing aerosols. Outside of the intrusion time window, the near surface was quite stratified, as is often the case in the background Arctic atmospheric state (classic stable boundary layer (Tjernström et al., 2019)). During the event, the near-surface both warms and becomes less stratified (i.e., weaker vertical gradient of equivalent potential temperature as a result of mixing). Part of this mixing is due to the aforementioned liquid-water clouds, which are optically thick and have strong cloud top radiative cooling, which drives buoyancy induced turbulent mixing of the atmosphere. The time evolution of the equivalent potential temperature indicates active mixing when the liquid-containing clouds are present, but not much mixing during the snowfall or background conditions. Together, the temperature gradients and FLEXPART simulations indicate high concentrations of pollutants transported at low-altitudes above the boundary layer from mid-latitudes, that are then subject to episodic vertical mixing that facilitates their transport into the boundary layer and their observation at near-surface levels. Altogether, these observations suggest that not only do aerosols impact clouds and their effects, but clouds can also impact the vertical distribution of aerosols.

3. I find some of the figures very hard to read. I understand that the authors want to use the same projection for each figure, however, if the observed signals are locally constrained (e.g., Figures 1d, S5, and S6) it is very difficult to detect the signal, especially the actual values. In Figure S6 the authors zoom in into the Kola Peninsula, which for example could be one option of emphasizing small-scale signals.

In the same figures it would also be helpful to remove the very small titles on top of each panel and label them as 'Background', 'First peak', and 'Second peak'.

We modified the figures as per the suggestion of the reviewer. The small titles are removed, the font is enlarged and inserts for figures 1d and S5 are now added in figure S5.

Specific comments

Abstract

Line 35: The potential effects on biogeochemistry are also mentioned in the ‘Implications’ section, but are not further discussed or explained. I would remove this here or replace it by ‘environmental properties’ (which would also include the effect on surface albedo, as briefly discussed in the manuscript) or something similar.

Modified

I know that the authors are limited in length, but in my opinion, it would be worth mentioning in the Abstract that the presented data are unique measurements from the remote central Arctic Ocean, given that aerosol measurements from the high Arctic are still sparse.

Modified. The abstract now reads:

Frequency and intensity of warm and moist air-mass intrusions into the central Arctic have increased over the past decades and have been related to sea ice melt. During the year-long MOSAiC expedition (Oct. 2019-Sept. 2020), which encompasses a set of unique measurements from the remote central Arctic Ocean, a record-breaking increase in temperature, moisture and downwelling longwave radiation were observed in mid-April 2020, during an air-mass intrusion carrying air pollutants from northern Eurasia. The two-day intrusion, caused the highest aerosol mass, particle number and cloud condensation nuclei number concentrations throughout the entire expedition-year, with drastic changes in the aerosol size distribution, chemical composition and particle hygroscopicity. The intrusion transformed the Arctic from a remote low-particle environment to an area comparable to a central-European urban setting. Additionally, such intrusion events, which are expected to intensify, resulted in an explosive increase in cloud condensation nuclei number, which can have direct effects on Arctic cloud properties in terms of radiation, precipitation patterns, and lifetime. Thus, unless prompt actions to significantly reduce emissions in the source regions are taken, such intrusion events are expected to continue to affect the Arctic climate and atmospheric properties.

Introduction

Line 51: It might be worth adding “nor quantified” as to my knowledge the measurements presented here are quite unique.

Modified

Lines 63-64: This statement is a little confusing, doesn’t the atmospheric circulation affect the synoptic weather conditions, not the other way around?

Modified

Same lines and following sentence: I would also argue that the chemical composition and properties also depend on the (local) emission sources.

The sentences are modified to: Local aerosol concentrations, their chemical composition and properties depend on atmospheric circulations, and are affected by season, local emissions and emissions far away that are atmospherically processed during long-range transport.

Line 67: To my understanding, Arctic haze is most prominent in spring (this is also stated in Freud et al., 2017 and others).

This is correct, and based on the most recent findings from multiple land-based observatories in the Arctic, Arctic haze extends between January and April (Schmale et al., 2022). Based on our observations of black carbon and accumulation mode particle concentrations during MOSAiC, haze is most prominent during January and February. This might be specific to the central Arctic ocean and to the year 2020, therefore we modified the sentence to include ‘early Spring’:

The Arctic aerosol annual cycle, as measured across Arctic land-based observatories (Schmale et al., 2022), is characterized by a dominant accumulation mode and high mass in winter/early spring (Arctic haze) and a dominant Aitken mode and low mass in summer.

Line 87: Related to my comment 2, in order to make this statement I think it would be worth showing some cloud property observations in the manuscript.

As per the reviewer’s suggestion, we included cloud observation from MOSAiC in Fig. 6.

Results

Lines 95-96/Figure S1: would it be possible to also show the intrusion as a vertical profile to see the vertical extent of the anomalies? Or was this temperature increase only seen at the surface?

The intrusion alters the temperature not only at near-surface but extends to the free troposphere. We added the following figure as part of Fig. 6 as well as a discussion on the vertical mixing triggered by the intrusion.

Lines 97-103: This reads more like an introduction and is a repetition, as air-mass intrusions have already been introduced in the introduction before (line 52ff), hence, I would suggest to remove this paragraph here.

We removed the paragraph.

Figure S2: Does the figure show surface (as indicated in the figure caption) or 2m (as indicated on top of the figure) temperature? Also, zooming in a bit into the Arctic (but still capturing the extent of the intrusion) would increase the visibility of the marked locations.

The reviewer is right, the figure shows the near-surface temperature. We added an insert to the figure to make it visible.

Line 108: Related to my previous comment, here it would be nice to have a vertical profile of temperature during the intrusion.

Added in Fig. 6. Please refer to the second general comment.

Figure 1D: What does the number on the lower right corner indicate?

The numbers indicate the total BC source contribution at the position of *Polarstern*. However, given that this number does not provide additional information, the numbers are removed to make the figure more-readable but kept and explained in figure S5.

Line 136: here the authors mention that the background aerosol concentration is representative of haze conditions, while in the introduction only winter is mentioned as the main period for Arctic haze. Please adjust this in the introduction.

Modified: *The Arctic aerosol annual cycle, as measured across Arctic land-based observatories (Schmale et al., 2022), is characterized by a dominant accumulation mode and high mass in winter/early spring (Arctic haze) and a dominant Aitken mode and low mass in summer.*

Line 160: the statement that the aerosol has been transformed during long-range transport is only discussed in the next section, hence, I would not make this statement here yet.

Modified: *The mass concentration observed during the intrusion period is extreme within the central Arctic Ocean context and demonstrates the large effect of mid-latitude pollution transported to the Arctic.*

Line 164: From the relative contributions reported in lines 173-174 and from Figures 3 and S7, the mass composition of the aerosol during the background period and the intrusion doesn't seem so different, rather the absolute mass/concentration is different. I would suggest slightly rewriting this paragraph, so this difference becomes clear.

In light of the reviewer's question, we improved our AMS analysis by homogenizing the April data with rest of the AMS data from the MOSAiC campaign. This resulted in a change in the calibration factor (relative ionization efficiency) of NH_4 and SO_4 , and hence a shift in the overall concentration. The improved concentrations fall much closer to the 1:1 line than before, when compared to the calculated mass concentration acquired from the scanning mobility particle sizer (Fig. R1), the changes in the concentrations do not affect any of the paper's conclusions but minor changes in some figures are needed. For reference the maximum mass concentration was $17.9 \mu\text{g}/\text{m}^3$ and now it is $13.8 \mu\text{g}/\text{m}^3$.

Figure R1 Comparison between PM_{10} mass concentration measured using the AMS and calculated from the volume distribution measured using the SMPS for April 2020.

In the updated version of Fig. 3, although sulfate still constitutes a major fraction of the aerosol mass, a higher contribution of organics, black carbon and ammonium is observed during the second peak of the intrusion relative to the first peak and the background period which highlights the difference in the source regions. We modified the sentence as follows:

The aerosol mass was dominated by sulfate throughout April (Fig. S7), yet, a strong variation in the mass composition of the aerosol during the intrusion was observed (Fig. 3).

What is also important to note is that while the bulk species composition might not change that significantly, the more detailed organic compound speciation does.

Same line: Why is not Figure 3 referenced here? I also don't really understand the added value of showing Figure S7 in addition to Figure 3, which is already very comprehensible. Given the large number of supplementary figures, I would consider removing Figure S7.

The reviewer is right, we now referenced Figure 3. We would like to keep figure S7 which gives an overview of the whole month of April, and the extent of change the intrusion introduced during these two days, in terms of mass. We also show the higher time resolution of PM_{10} , which is essential for the previous section of the manuscript.

Lines 190-195: I don't really understand why these peaks would not have been observed at other Arctic observatories, but on the *Polarstern*. Could the authors elaborate on the reason for this? Currently, I find this paragraph a little confusing.

Air mass transport was such that the land-based observatories were either not affected at all (e.g., Villum) or only marginally (e.g., Zeppelin). We show in Fig. R2 that SO_4 measured from aerosol collected over several days on filter samples at Zeppelin does not show any signature of the intrusion (Tørseth et al., 2012). Higher resolution aerosol data shows some signature (personal communication with Paul Zieger, Stockholm University).

Figure R2 Sulfate in the aerosol phase at Zeppelin Observatory. The warm air-mass intrusion event discussed in this manuscript is within the unshaded area. Data available from EBAS (<http://ebas-data.nilu.no/>, last access: 29.04.2022).

We modified the paragraph to:

Interestingly, neither a noteworthy peak in SO_4^{2-} nor in SO_2 was observed at Zeppelin, Svalbard during the intrusion event on April 15th – 16th (Fig. S5), although the location was affected by the intrusion in terms of temperature (Fig S2). It appears that while the advection of temperature occurs over a broad swath, the advection of aerosols/pollution happens over narrower parts of the overall event due to specific point sources embedded within the general source region. Such a limited observation of the intrusion in certain land-based Arctic locations (here, Zeppelin) and their absence at others (e.g., VRS, Greenland), demonstrates the importance of in-situ central Arctic Ocean measurements as land-based observations cannot be simply extrapolated.

Line 223: Are these correlations significant, i.e., did the authors perform a significance test?

The correlations are indeed significant, we added the p-values to the figure S10 and to the sentence on line 223 which now reads:

Conversely, during the warm intrusion period the correlation between the two species increases substantially with $R^2 = 0.44$ ($p\text{-value} = 3.8 \times 10^{-4}$) and $R^2 = 0.70$ ($p\text{-value} = 3.9 \times 10^{-7}$), during the 1st and 2nd peak, respectively. The high significant correlation is an indication of the common transport of BC and SO_4^{2-} from the source region to the central Arctic.

Figure S10, caption (line 1058): again, did the authors perform a significance test? If not, I would rephrase “significant difference”.

We added the p-values to the figure.

Figure 5: It might be helpful to rename “all data” to “whole expedition” or similar, as in the previous figures “all data” referred to April 2020 and not the whole year.

We thank the reviewer for pointing this out. We modified ‘all data’ to ‘MOSAIC year’.

Line 390: increased CCN concentrations do not necessarily lead to an increase in net surface longwave warming, in fact, Eirund et al. (2019) showed that low-level clouds over sea ice perturbed by CCN become more optically thick, however, the effect in net surface LW remains small. There, the change in cloud structure might be responsible for the radiative response of the clouds to CCN concentrations changes, as stratocumulus clouds over the ocean show a larger response in net surface LW. Note that in Eirund et al., 2020 no CCN perturbations were applied, so this reference might be a bit confusing here. The radiative response in the low-level clouds in this study was solely driven by the temperature and moisture perturbations.

We agree with the reviewer, please refer to point 2, under general comments.

Implication

Line 406: I would rather show evidence in form of cloud observations here or (if available) cite publications related to this event.

Added.

Line 428: why extreme conditions?

We meant during intrusion periods. The sentence now reads:

Therefore, an expected increase in the frequency of intrusions together with policy-imposed decreases in SO_2 emissions in Russia and elsewhere, might point towards a rather substantial decrease in central Arctic aerosol acidity.

Line 440: consider citing some other studies here which have looked at the CCN impact in Arctic clouds

We added the following citations:

While the net impact of increased CCN concentrations in this environment involves a complex web of cloud microphysical processes (Lubin and Vogelmann, 2006; Stevens et al., 2018), such an increase could mean that optically thicker clouds can form over the Arctic pack ice leading to strong downwelling longwave radiation (Eirund et al., 2020), which was observed to be abnormally high in this case (Rinke et al., 2021), thereby positively re-enforcing the warming effect.

Line 444: “biogeochemical implications” has been mentioned before, but what specifically is meant here? Ocean biogeochemical changes?

By “biogeochemical” we refer to the cycling of elements between the ocean, atmosphere and biosphere, for example nitrogen. The enhanced nitrogen transport in the form of ammonium during the warm air-mass intrusion is one aspect. We modified the sentence:

As of yet, there are no such comprehensive central Arctic Ocean aerosol observations and land-based observatories only partially capture such intrusion events, underlining the uniqueness of our measurements, and the need to more deeply understand climate and biogeochemical implications (i.e. transport of nitrogen to the central Arctic).

Line 452: “short and long-time scales” is there any evidence of a long-term effect of aerosol injections on Arctic cloud microphysics from either the available measurements or previous studies?

The reviewer brings up a very interesting point here. In reference to general comments point 2, we believe that the increased CCN concentration may have longer-time scale effects on the Arctic clouds, affecting precipitation patterns and clouds’ lifetime. However, there are no long-term in-situ observations that serve to answer such an important research question. To know whether long-term effects exist, modeling experiments are needed. This is indeed a very exciting research question that deserves its own effort.

We modified the sentence to avoid any speculations:

It is however clear that pulse-injections of pollution into the Arctic in terms of aerosol number concentration, climate relevant compounds such as sulfate, organics and BC, as well as possibly environmental pollutants (which we did not measure), modify Arctic cloud microphysics and the chemical regime drastically such that these need to be taken into account in chemistry transport and climate models.

Methods

Line 515: Was MOSAiC really the first year-round expedition in the Arctic? To my knowledge, SHEBA was the first (and until MOSAiC the only) one (e.g., Shupe et al., 2006)

The reviewer is right, indeed MOSAiC was not the first year-round expedition but the most comprehensive, reaching the central Arctic Ocean and in terms of aerosol measurements. The sentence now reads:

The MOSAiC (Multidisciplinary drifting Observatory for the Study of Arctic Climate) expedition is the most comprehensive year-round expedition into the central Arctic exploring the Arctic climate system.

Line 526: Do the authors mean “heated”?

Yes, modified. Thank you.

References

Eirund et al., 2019: Response of Arctic mixed-phase clouds to aerosol perturbations under different surface forcing, Atmos. Chem. Phys., 19, <https://doi.org/10.5194/acp-19-9847-2019>

Kanji et al., 2017: Overview of Ice Nucleating Particles, Met. Monographs, 58, <https://doi.org/10.1175/AMSMONOGRAPHS-D-16-0006.1>

Shupe et al., 2006: Arctic Mixed-Phase Cloud Properties Derived from Surface-Based Sensorsat SHEBA, Journal of Atmospheric Sciences, 63, <https://doi.org/10.1175/JAS3659.1>

Stevens et al., 2019: A Model Intercomparison of CCN-Limited Tenuous Clouds in the High Arctic, *Atmos. Chem. Phys.*, 18, <https://doi.org/10.5194/acp-18-11041-2018>

Reviewer #2

Review for „A Central Arctic Extreme Aerosol Event Triggered by a Warm Air-Mass Intrusion“ by Dada et al. submitted to Nature Communications

Summary & Recommendation

This study provides an in-depth characterization of aerosol composition and air mass history during the strongest warm air mass intrusion observed during the MOSAiC field campaign in the high Arctic. The intrusion was characterized by anomalously warm and moist near-surface air and significantly higher aerosol concentrations for a duration of about two days (15.-16.04.2020). Furthermore, these particles were likely considerably „younger“ (<2 days) than average spring time Arctic haze particles observed during the reference period (02.-03.04.). Furthermore, different aerosol compositions and size distributions were observed and aerosols were characterised by different hygroscopicity and acidity. All of these changes may impact aerosol-cloud -, and aerosolradiation interactions, as well as atmospheric chemistry and secondary organic nucleation rates. Currently our understanding of the aerosol impact of air mass intrusions into the high Arctic, their impact on the local biogeochemistry and aggregated importance (i.e. across several events) under climate change and Arctic amplification remains unclear. Dada et al. show an in-depth analysis of such an event which was missed by ground stations of similar latitude using a wide array of instruments available during the MOSAiC campaign. This in itself highlights the issues with sampling these events in a region where ground-based observations are sparse.

The study is novel, well-conceived, well written, and very carefully analysed. It is the first analysis of this kind presented in the scientific literature and I recommend publication in Nature Communications once the minor points mainly regarding clarifications are addressed.

We thank the reviewer for their thoughtful comments that help improve our manuscript. We revised the manuscript following the reviewers' suggestions and responded to the point-by-point comments in blue. Additions to the text are shown in *italics*.

Minor Revisions

L36: “[...],but are expected to intensify” I personally find this statement too strong. While the changes in aerosol concentrations and their impact on chemistry, clouds and radiation within the warm air intrusion is undoubtedly significant, these events (as you also state) are short-lived. Prolonged impacts or aggregate impacts across multiple intrusions on long time scales are not yet known. It could be thus be that while these events occur more often, the aerosol-driven effects associated with warm-air intrusions are insignificant on longer time scales.

We agree with the reviewer that the longer time scales of the warm intrusions' effects are still unknown, we intended to say that the warm intrusions are expected to intensify based on future projections. We modified the sentence as follows:

Additionally, such intrusion events, which are expected to intensify, resulted in an explosive increase in cloud condensation nuclei number, which can have direct effects on Arctic cloud properties in terms of radiation, precipitation patterns, and lifetime. Thus, unless prompt actions to significantly reduce emissions in the source regions are taken, such intrusion events are expected to continue to affect the Arctic climate and atmospheric properties.

We also refer the reviewer to the new paragraph in the implication section and Fig. 6, where we discuss potential longer effects of the aerosol-cloud interactions, that is beyond the time horizon of our observations.

L39-L87: The introduction is very well written and comprehensive. My only suggestion would be to consider to incorporate 1-2 sentences on alternate approaches in characterizing intrusions in terms of their aerosol change in terms of AOD using remote sensing. E.g.: <https://acp.copernicus.org/preprints/acp-2021-805/> and references therein.

As per suggestion from the reviewer, the last paragraph of the introduction now reads:

Measurements of aerosol optical properties in the Arctic provide high spatial and long-term climatological information important for model simulations, however remain mainly site-specific and do not often extend fully to the central Arctic Ocean (Xian et al., 2021; Glantz et al., 2014; Engelmann et al., 2021). Additionally, while land-based observatories, provide climatological information about Arctic particle size distribution and mass composition that is highly valuable for understanding Arctic aerosol processes and for modeling their climate effects (Schmale et al., 2021), they are not necessarily representative for aerosol characteristics and impacts over the remote central Arctic Ocean. The effects of warm air-mass intrusions on the particle population have not yet been studied over the central Arctic Ocean, because in-situ measurements were previously only conducted in summer (Heintzenberg et al., 2015; Chang et al., 2011). The year-long MOSAiC expedition (Multidisciplinary drifting Observatory for the Study of Arctic Climate; Oct. 2019-Sept. 2020) enabled scientists to study the central Arctic's atmosphere in detail (Shupe et al., 2020; Shupe et al., 2022). During the expedition, a record minimum sea ice extent was measured in July 2020 (compared to 1979-2020) (Liang et al., 2021). This major sea ice retreat was preceded by a record-breaking air-mass intrusion, which was high in moisture, temperature and longwave radiation re-emitted by low level clouds in the Arctic, all being the highest in the last 40 years (Rinke et al., 2021). This warm and moist air-mass intrusion event was observed in April 2020 and is the focus of this study. Our unique, in-situ, measurements in the remote central Arctic Ocean demonstrate the capability of pollutants emitted in lower latitudes to drastically alter the Arctic atmosphere, one of the world's most climate sensitive locations.

Fig 1: Generally it is not clear to me what it means to plot a variable "along the emission sensitivity". Emission sensitivity is defined in L653ff as the impact of a unit of emission on the tracer concentration at the ship. So are you looking along a gradient of emission sensitivities and what is the range of this gradient: 0-1? Generally do you consider all air masses with an emission sensitivity larger than 0, or do you use another threshold? What does it mean to show temperature at 1500m along the emission sensitivity? In the caption you write that its for emission sensitivities below 1500m. Is that in the same column? Or do you mean the air mass had previously been near the surface with a layer of an emission sensitivity >0? For me some more detail in the text is needed here to really understand how you use this concept.

We agree that the expression "along the emission sensitivity" can be misleading. What is meant is that we show the temperature of the airmasses in the regions where the emission sensitivity is larger than zero, i.e. the regions from where we have advection.

We hence reformulate the caption as: (C) Mean temperature of the air mass with simulated particles residing below 100 m altitude, obtained from 7-day backward simulations with FLEXPART during the background period (left panel), first peak of the intrusion event (middle panel) and second peak of the intrusion event (right panel). (D) Black carbon source contribution (below 100 m a.g.l.), as a surrogate for anthropogenic pollution, for a passive air tracer obtained from 7-day backward simulations with FLEXPART during the background period (left panel), first peak of the intrusion event (middle panel) and second peak of the intrusion event (right panel). Sea ice concentration (in %) is shown as grey scales. (E) Average altitude of all particles residing below 1500 m, from the 7-day FLEXPART backward calculation for the background period (left panel), first peak of the intrusion event (middle panel) and second peak of the intrusion event (right panel). In all of (C), (D) and (E) the position of Polarstern is marked with a star, that of Norilsk with a square, that of Vorkuta with a triangle and that of Murmansk with a circle.

Fig1c: Why do you show these maps at 1500m altitude? This is above the warm air intrusion as you state in line 108 and considerably above the BL, which the emissions need to be mixed into.

The reviewer is right. We replaced the maps in Fig. 1C with the ones at 100 m.

Generally it would be nice to see a vertical cross section along the warm air intrusion in addition to the horizontal slices. This would shed light on its vertical structure and help understanding how exactly the polluted air mass is in contact with the BL. This doesn't have to be along the emission sensitivity filtered

fields, but could simply show the thermodynamic structure as its shown in Fig S2 in terms of its horizontal extend.

Please see the new figure 6 and discussion added in the implications section (as requested by both reviewers). We show in Fig. 6b the vertical profile of the equivalent potential temperature which shows the vertical extent and boundary layer stability during the event.

Fig1: 3rd columns. Rows C and E show these ripple structures along the temperature and altitude. Can you explain these?

The structures that resemble "ripples" are due to the time resolution of the model output, three hours. Therefore, the "discontinuities" shown are due to the displacement undergone by the air masses in that time interval. We added this info to the figure caption.

Fig1 caption: "Altitude profile of the emission sensitivities shown in panel (C)" Revise wording. Panel C does not show emission sensitivity, but temperature fields along the emission sensitivity.

Modified.

L240ff: During the first peak the correlation is similarly low. Can you explain this? I assume its due to the different source regions and resulting differences in composition?

This is correct. The weak correlation could be attributed to the mixture of sources forming the 'background' Arctic haze. We modified the sentence as follows:

During the background period, no clear connection between SO_4^{2-} and BC is apparent (Fig. S8A), likely as a result of their low concentrations but also suggesting a broad mix of sources that cannot be back-traced to individual emission regions or sources.

L419ff: You state that aerosol-radiation interactions may be important. Could you not diagnose these during the event using ground-based measurements on radiation obtained during MOSAiC?

As per the recommendation of both reviewers, we added cloud observations from *Polarstern* to visualize the impact of the intrusion on the central Arctic atmosphere (Fig. 6 in the main text). Here, we included the radar reflectivity to show the cloud height and the clouds' liquid water path. During the air-mass intrusion, we observe mixed-phase clouds that form near the surface and up around 2 km height, which could be impacted by CCN concentration. During the start of the intrusion period, deep ice-clouds are observed, which precipitate. Additionally, the intrusion period is interrupted by heavy snow fall. Both of these precipitating periods inhibit the formation of liquid clouds at lower levels (seeder-feeder mechanism). When significant deep precipitation is not occurring, classic Arctic mixed-phase stratiform clouds are present. The observed CCN increase with the warm air advection could play an important role for those clouds and for providing a source of new aerosols/CCN for the following "background state" of the Arctic (i.e., the increased CCN concentration will remain with this advected air mass for an extended time downstream). We also included the up and downwelling longwave radiation (Fig. 6d) as well as the net longwave radiation (Fig. 6e). The net LW shows the increased opacity of the clouds during the intrusion period associated with lots of cloud particles and liquid-water containing clouds.

Also, as reviewer 1 pointed out, the impact of increasing CCN on downwelling longwave radiation might not be that large in this case because the background concentration is already pretty high given that the clouds are already optically thick ($LWP > 30 \text{ g/m}^2$). In this case, there is little change in cloud emissivity because the cloud is already effectively a blackbody emitter. There might be larger impacts on the shortwave radiation, which reaches its "saturation" point at a higher cloud LWP. Also, in addition to these radiative effects, there are other potential implications of increased CCN during these events:

(1) Precipitation: for the liquid containing clouds, a much higher CCN concentration leads to more numerous, smaller droplets. These are significantly less effective at growing to precipitation sizes. Additionally, these smaller droplets are expected to be less effective at forming ice (Lance et al., 2004;Norgren et al., 2018). Thus, this higher concentration of CCN would mean less precipitation from the mixed-phase clouds and therefore a weakened precipitation sink of the atmospheric moisture.

(2) Lifetime: Building on that point, the increase in CCN could impact the time scales associated with the air-mass transformation. With less efficient loss of moisture, the clouds would last longer than they otherwise would have. That lifetime is related to both the total moisture budget and the CCN budget. So presumably the clouds would last much longer before they become limited by moisture or particles. As a result of this increased cloud lifetime, the cloud radiative effects would also last longer and thus have an impact over a longer time and larger spatial scale.

By including in addition the vertical temperature structure of the atmosphere during the intrusion, another characteristic of the intrusion is revealed, vertical mixing. We find that outside of the intrusion time window, the near surface was quite stratified, as is often the case in the background Arctic atmospheric state (classic Arctic stable boundary layer). During the event, the near-surface both warms and becomes less stratified (i.e., more mixing). Part of this mixing is due to the aforementioned liquid-water clouds, which are optically thick, and have strong radiative cooling to space, driving turbulent mixing of the atmosphere. The temporal transition in equivalent potential temperature vertical gradients (Fig. 6b) shows how active mixing occurred during the intrusion periods, compared to not much mixing during the strong snowfall or background conditions. Overall, the temperature gradients and FLEXPART simulations indicate high concentrations of pollutants transported at low-altitudes above the boundary layer from mid-latitudes, that are then subject to episodic vertical mixing that facilitates their transport into the boundary layer and their observation at near-surface levels. The liquid water clouds present during the intrusion period, and specifically their cloud top radiative cooling that drives buoyant circulations over some depth, likely played a major role in mixing the aerosol down towards the surface.

We added the following figure and associated discussion to the ‘Implications’ section:

To visualize the effect of the increased CCN concentration on the Arctic clouds, we include cloud observations from Polarstern (Fig. 6). Radar reflectivity measurements show deep ice-clouds at the beginning of the intrusion period, as well as periodic snowfall events before, throughout and after the intrusion period (i.e., reflectivity > 0 dBZ reaching the surface). During the intrusion event, when deeper precipitation is not present, stratiform mixed-phase clouds form near the surface and up to about 2 km height. At the same time, the up and downwelling longwave radiation (Fig. 6d) show the increased opacity of the clouds during the intrusion period associated with the occurrence of liquid-water containing clouds or deep ice clouds. Here, the CCN increase could play an important role for the liquid-containing clouds both through direct radiative effects and by supplying this air-mass with a high CCN concentration to impact cloud processes downstream. Given the already high CCN concentration during the Arctic ‘background state’, i.e. haze period, and the fact that the liquid-containing clouds are already optically thick (liquid water path > 30 g/m², panels c-e), the impact of increased CCN on downwelling longwave radiation might not be that large (Eirund et al., 2019) at the time the air-mass passed over Polarstern. Yet, the increased CCN concentrations might have larger impacts on the cloud reflection of shortwave radiation via the Twomey effect (Twomey, 1974), which occurs over a wider range of liquid water path values. In addition to these instantaneous radiative effects, increased CCN concentrations during such warm intrusion events could affect the precipitation and lifetime of liquid containing clouds in the Arctic. A much higher CCN concentration leads to more numerous, smaller droplets, which are significantly less effective at growing to precipitation sizes. Additionally, such smaller droplets are expected to be less effective at forming ice (Lance et al., 2004;Norgren et al., 2018). Thus, this high concentration of CCN would inhibit precipitation from the

mixed-phase clouds and therefore weaken their primary sink of moisture. Building on that point, the increase in CCN, based on our case study, could be impacting the time scales associated with the air-mass transformation. With less efficient loss of moisture and more plentiful CCN, the air-mass could sustain clouds for longer than it otherwise would have before moisture or CCN availability became a limiting factor. Overall, this potential extension of the cloud lifetime means that the cloud radiative effects would occur over a longer time and larger spatial extent during the progression of the intrusion event.

In addition, the vertical equivalent potential temperature structure of the atmosphere during the intrusion (Fig. 6b) reveals the role of vertical mixing for dispersing aerosols. Outside of the intrusion time window, the near surface was quite stratified, as is often the case in the background Arctic atmospheric state (classic stable boundary layer (Tjernström et al., 2019)). During the event, the near-surface both warms and becomes less stratified (i.e., weaker vertical gradient of equivalent potential temperature as a result of mixing). Part of this mixing is due to the aforementioned liquid-water clouds, which are optically thick and have strong cloud top radiative cooling, which drives buoyancy induced turbulent mixing of the atmosphere. The time evolution of the equivalent potential temperature indicates active mixing when the liquid-containing clouds are present, but not much mixing during the snowfall or background conditions. Together, the temperature gradients and FLEXPART simulations indicate high concentrations of pollutants transported at low-altitudes above the boundary layer from mid-latitudes, that are then subject to episodic vertical mixing that facilitates their transport into the boundary layer and their observation at near-surface levels. Altogether, these observations suggest that not only do aerosols impact clouds and their effects, but clouds can also impact the vertical distribution of aerosols.

Fig2: Do you have a hypothesis for why the accumulation mode leads the Aitken mode during the second peak?

The reviewer raises a very interesting point which might merit its own future research. Given that our observation was fixed to a specific location (*Polarstern*), while the air-mass was not stagnant, the observation of the accumulation mode prior to that of the Aitken mode might be related to the movement of the air-mass. What we observed was precipitation between the two intrusion peaks, which removed much of the accumulation mode. It “recovers” more quickly than the intense Aitken mode at the end of the observation period. This likely points to the characteristics of particles in the moving air mass, rather than on site formation of either or both modes. Additionally, focusing at the end of the observation period with the strong Aitken mode, if precipitation occurred along the trajectory, the accumulation mode could have been washed out over time, while the Aitken mode preferentially remains and has a chance to grow. Using a box model along a Lagrangian trajectory by which we follow the air parcel from its emission source to the central Arctic Ocean would be ideal, but not easy to implement and evaluate given the unavailability of multiple simultaneous observations in the high Arctic.

L661: ECLIPSE: please write out acronym
Added

Edits

L40: Arctic amplification in my opinion is a phenomenon, not a process.
Modified

L50: “and the related” -> “and related”
Modified

L106: double spacing
Modified

L702: missing space
Modified

References

- Chang, R. Y. W., Leck, C., Graus, M., Müller, M., Paatero, J., Burkhardt, J. F., Stohl, A., Orr, L. H., Hayden, K., Li, S. M., Hansel, A., Tjernström, M., Leaitch, W. R., and Abbatt, J. P. D.: Aerosol composition and sources in the central Arctic Ocean during ASCOS, *Atmos. Chem. Phys.*, 11, 10619-10636, 10.5194/acp-11-10619-2011, 2011.
- Creamean, J., Hill, T., DeMott, P., Barry, K., and Hume, C.: Arctic Ice Nucleation Sampling during MOSAiC (INPMOSAIC2) Field Campaign Report, Oak Ridge National Lab.(ORNL), Oak Ridge, TN (United States). ARM user facility, 2021.
- Eirund, G. K., Possner, A., and Lohmann, U.: Response of Arctic mixed-phase clouds to aerosol perturbations under different surface forcings, *Atmos. Chem. Phys.*, 19, 9847-9864, 10.5194/acp-19-9847-2019, 2019.
- Eirund, G. K., Possner, A., and Lohmann, U.: The Impact of Warm and Moist Air Mass Perturbations on Arctic Mixed-Phase Stratocumulus, *Journal of Climate*, 33, 9615-9628, 10.1175/jcli-d-20-0163.1, 2020.
- Engelmann, R., Ansmann, A., Ohneiser, K., Griesche, H., Radenz, M., Hofer, J., Althausen, D., Dahlke, S., Maturilli, M., Veselovskii, I., Jimenez, C., Wiesen, R., Baars, H., Bühl, J., Gebauer, H., Haarig, M., Seifert, P., Wandinger, U., and Macke, A.: Wildfire smoke, Arctic haze, and aerosol effects on mixed-phase and cirrus clouds over the North Pole region during MOSAiC: an introduction, *Atmos. Chem. Phys.*, 21, 13397-13423, 10.5194/acp-21-13397-2021, 2021.
- Glantz, P., Bourassa, A., Herber, A., Iversen, T., Karlsson, J., Kirkevåg, A., Maturilli, M., Seland, Ø., Stebel, K., Struthers, H., Tesche, M., and Thomason, L.: Remote sensing of aerosols in the Arctic for an evaluation of global climate model simulations, *Journal of Geophysical Research: Atmospheres*, 119, 8169-8188, <https://doi.org/10.1002/2013JD021279>, 2014.
- Heintzenberg, J., Leck, C., and Tunved, P.: Potential source regions and processes of aerosol in the summer Arctic, *Atmos. Chem. Phys.*, 15, 6487-6502, 10.5194/acp-15-6487-2015, 2015.
- Lance, S., Nenes, A., and Rissman, T. A.: Chemical and dynamical effects on cloud droplet number: Implications for estimates of the aerosol indirect effect, *Journal of Geophysical Research: Atmospheres*, 109, <https://doi.org/10.1029/2004JD004596>, 2004.
- Liang, Y., Bi, H., Huang, H., Lei, R., Liang, X., Cheng, B., and Wang, Y.: Warm and moist atmospheric flow caused a record minimum July sea ice extent of the Arctic in 2020, *The Cryosphere Discuss.*, 2021, 1-23, 10.5194/tc-2021-159, 2021.
- Lubin, D., and Vogelmann, A. M.: A climatologically significant aerosol longwave indirect effect in the Arctic, *Nature*, 439, 453-456, 10.1038/nature04449, 2006.
- Norgren, M. S., de Boer, G., and Shupe, M. D.: Observed aerosol suppression of cloud ice in low-level Arctic mixed-phase clouds, *Atmos. Chem. Phys.*, 18, 13345-13361, 10.5194/acp-18-13345-2018, 2018.
- Rinke, A., Cassano, J. J., Cassano, E. N., Jaiser, R., and Handorf, D.: Meteorological conditions during the MOSAiC expedition: Normal or anomalous?, *Elementa: Science of the Anthropocene*, 9, 10.1525/elementa.2021.00023, 2021.
- Schmale, J., Sharma, S., Decesari, S., Pernov, J., Massling, A., Hansson, H. C., von Salzen, K., Skov, H., Andrews, E., Quinn, P. K., Upchurch, L. M., Eleftheriadis, K., and Traversi, R.: Pan-Arctic seasonal cycles and long-term trends of aerosol properties from ten observatories, *Atmos. Chem. Phys. Discuss.*, 2021, 1-53, 10.5194/acp-2021-756, 2021.

Schmale, J., Sharma, S., Decesari, S., Pernov, J., Massling, A., Hansson, H. C., von Salzen, K., Skov, H., Andrews, E., Quinn, P. K., Upchurch, L. M., Eleftheriadis, K., Traversi, R., Gilardoni, S., Mazzola, M., Laing, J., and Hopke, P.: Pan-Arctic seasonal cycles and long-term trends of aerosol properties from 10 observatories, *Atmos. Chem. Phys.*, 22, 3067-3096, 10.5194/acp-22-3067-2022, 2022.

Shupe, M. D., Rex, M., Dethloff, K., Damm, E., Fong, A. A., Gradinger, R., Heuzé, C., Loose, B., Makarov, A., Maslowski, W., Nicolaus, M., Perovich, D., Rabe, B., Rinke, A., Sokolov, V., and Sommerfeld, A.: Arctic Report Card 2020: The MOSAiC Expedition: A Year Drifting with the Arctic Sea Ice, <https://doi.org/10.25923/9g3v-xh92>, 2020.

Shupe, M. D., Rex, M., Blomquist, B., Persson, P. O. G., Schmale, J., Uttal, T., Althausen, D., Angot, H., Archer, S., Bariteau, L., Beck, I., Bilberry, J., Bucci, S., Buck, C., Boyer, M., Brasseur, Z., Brooks, I. M., Calmer, R., Cassano, J., Castro, V., Chu, D., Costa, D., Cox, C. J., Creamean, J., Crewell, S., Dahlke, S., Damm, E., de Boer, G., Deckelmann, H., Dethloff, K., Dütsch, M., Ebell, K., Ehrlich, A., Ellis, J., Engelmann, R., Fong, A. A., Frey, M. M., Gallagher, M. R., Ganzeveld, L., Gradinger, R., Graeser, J., Greenamyre, V., Griesche, H., Griffiths, S., Hamilton, J., Heinemann, G., Helmig, D., Herber, A., Heuzé, C., Hofer, J., Houchens, T., Howard, D., Inoue, J., Jacobi, H.-W., Jaiser, R., Jokinen, T., Jourdan, O., Jozef, G., King, W., Kirchgaessner, A., Klingebiel, M., Krassovski, M., Krumpfen, T., Lampert, A., Landing, W., Laurila, T., Lawrence, D., Lonardi, M., Loose, B., Lüpkes, C., Maahn, M., Macke, A., Maslowski, W., Marsay, C., Maturilli, M., Mech, M., Morris, S., Moser, M., Nicolaus, M., Ortega, P., Osborn, J., Pätzold, F., Perovich, D. K., Petäjä, T., Pilz, C., Pirazzini, R., Posman, K., Powers, H., Pratt, K. A., Preußner, A., Quéléver, L., Radenz, M., Rabe, B., Rinke, A., Sachs, T., Schulz, A., Siebert, H., Silva, T., Solomon, A., Sommerfeld, A., Spreen, G., Stephens, M., Stohl, A., Svensson, G., Uin, J., Viegas, J., Voigt, C., von der Gathen, P., Wehner, B., Welker, J. M., Wendisch, M., Werner, M., Xie, Z., and Yue, F.: Overview of the MOSAiC expedition: Atmosphere, *Elementa: Science of the Anthropocene*, 10, 10.1525/elementa.2021.00060, 2022.

Stevens, R. G., Loewe, K., Dearden, C., Dimitrellos, A., Possner, A., Eirund, G. K., Raatikainen, T., Hill, A. A., Shipway, B. J., Wilkinson, J., Romakkaniemi, S., Tonttila, J., Laaksonen, A., Korhonen, H., Connolly, P., Lohmann, U., Hoose, C., Ekman, A. M. L., Carslaw, K. S., and Field, P. R.: A model intercomparison of CCN-limited tenuous clouds in the high Arctic, *Atmos. Chem. Phys.*, 18, 11041-11071, 10.5194/acp-18-11041-2018, 2018.

Tjernström, M., Shupe, M. D., Brooks, I. M., Achtert, P., Prytherch, J., and Sedlar, J.: Arctic Summer Airmass Transformation, Surface Inversions, and the Surface Energy Budget, *Journal of Climate*, 32, 769-789, 10.1175/jcli-d-18-0216.1, 2019.

Tørseth, K., Aas, W., Breivik, K., Fjæraa, A. M., Fiebig, M., Hjellbrekke, A. G., Lund Myhre, C., Solberg, S., and Yttri, K. E.: Introduction to the European Monitoring and Evaluation Programme (EMEP) and observed atmospheric composition change during 1972–2009, *Atmos. Chem. Phys.*, 12, 5447-5481, 10.5194/acp-12-5447-2012, 2012.

Twomey, S.: Pollution and the planetary albedo, *Atmospheric Environment* (1967), 8, 1251-1256, 1974.

Xian, P., Zhang, J., Toth, T. D., Sorenson, B., Colarco, P. R., Kipling, Z., O'Neill, N. T., Hyer, E. J., Campell, J. R., Reid, J. S., and Ranjbar, K.: Arctic spring and summertime aerosol optical depth baseline from long-term observations and model reanalyses, with implications for the impact of regional biomass burning processes, *Atmos. Chem. Phys. Discuss.*, 2021, 1-63, 10.5194/acp-2021-805, 2021.